

# Two decades of satellite observations of AOD over mainland China.

Gerrit de Leeuw[1*], Larisa Sogacheva[1], Edith Rodriguez[1], Konstantinos Kourtidis[2], Aristeidis K. Georgoulias[2], Georgia Alexandri[2], Vassilis Amiridis[3], Emmanouil Proestakis[3,4], Eleni Marinou[3,5], Yong Xue[6], Ronald van der A[7]

[1]Finnish Meteorological Institute(FMI), Climate Research Unit, Helsinki, Finland

[2]Laboratory of Atmospheric Pollution and Pollution Control Engineering of Atmospheric Pollutants, Department of Environmental Engineering, Democritus University of Thrace, Xanthi, Greece

[3] National Observatory Athens  (NOA), Greece

[4]Laboratory of Atmospheric Physics, Department of Physics, University of Patras, 26500, Greece

[5]Department of Physics, Aristotle University of Thessaloniki, Thessaloniki, 54124, Greece

[6] Department of Electronics, Computing and Mathematics, College of Engineering and Technology, University of Derby, Derby DE22 1GB, UK

[7] Royal Netherlands Meteorological Institute (KNMI), De Bilt, Netherlands

* Correspondence to: Gerrit de Leeuw (gerrit.leeuw@fmi.fi)

**Abstract.** The retrieval of aerosol properties from satellite observations provides their spatial distribution over a wide area in cloud-free conditions. As such, they complement ground-based measurements by providing information over sparsely instrumented areas, albeit that significant differences may exist in both the type of information obtained and the temporal information from satellite and ground-based observations. In this paper, information from different types of satellite-based instruments is used to provide a 3-D climatology of aerosol properties over mainland China, i.e. vertical profiles of extinction coefficients from CALIOP, a lidar flying on board the CALIPSO satellite, and the column-integrated extinction (AOD), available from three radiometers: ESA's ATSR-2, AATSR (together referred to as ATSR) and NASA's MODIS/Terra, together spanning the period 1995-2015. AOD data are retrieved from ATSR using the ADV v2.31 algorithm while for MODIS the Collection 6 (C6) DTDB merged AOD data set is used. These data sets are validated and differences are compared using AERONET version 2 L2.0 AOD data as reference. The results show that, over China, MODIS slightly overestimates the AOD and ATSR slightly underestimates the AOD. Consequently, MODIS AOD is overall higher than that from ATSR, and the difference increases with increasing AOD. The comparison also shows that none of the ATSR and MODIS AOD data sets is better than the other one everywhere. However, ATSR ADV has limitations over bright surfaces where the MODIS DB was designed for. To allow for comparison of MODIS C6 results with previous analyses where MODIS Collection 5.1 (C5.1) data were used, also the difference between the C6 and C5.1 DTDB merged data sets from MODIS/Terra over China is briefly discussed.

The AOD data sets show strong seasonal differences and the seasonal features vary with latitude and longitude across China. Two-decadal AOD time series, averaged over the whole mainland China, are presented and briefly discussed. Using the 17 years of ATSR data as the basis and MODIS/Terra to follow the temporal evolution in recent years when ENVISAT was lost requires a comparison of the data sets for the overlapping period to show their complementarity. ATSR precedes the MODIS time series between 1995 and 2000 and shows a distinct



increase in the AOD over this period. The two data series show similar variations during the overlapping period between 2000 and 2011, with minima and maxima in the same years. MODIS extends this time series beyond the end of the ENVISAT period in 2012, showing decreasing AOD.

## 1. Introduction

An aerosol is a suspension of droplets and/or particles in a fluid (Seinfeld and Pandis, 1997). For atmospheric aerosols the fluid is the air and the aerosols are generally referred to as particles. This convention will also be followed in this manuscript. The particles usually consist of solid and / or liquid material, or a mixture of these, depending on their origin and aggregation state, dissolved in liquid, usually water. These particles can be more or less hygroscopic, depending on their chemical composition, and water vapour is released or taken up depending

further on the ambient relative humidity (RH), until they are in an equilibrium state. At very high RH, around 100%, hygroscopic particles are activated to cloud condensation nuclei, grow into the cloud droplet size range and are no longer considered aerosol particles. When they are transported into a lower RH environment the water evaporates but the remaining aerosol particle may have a different chemical composition than the original one. The initial chemical composition of an aerosol particle depends on the original generation mechanism, i.e. on the

sources such as biomass burning, dust lifted up from the surface or sea spray aerosol generated at the sea surface by the action of wind and waves. Alternatively, aerosol particles are generated from their precursor gases by nucleation under the influence of UV radiation or catalysts (e.g. Kulmala and Kerminen, 2008) after which these very small particles, with a radius of a few nm, may grow by condensation and coagulation. In this process their chemical composition may change and hence the atmospheric aerosol is a complex mixture of chemical

components distributed over a wide range of sizes spanning 5-6 orders of magnitude from a few nm to tens of μm, and a wide range of concentrations spanning ca. 10 orders of magnitude, depending on the particle size. Aerosol particles are important because of their effects on climate, health, atmospheric chemistry, visibility, cultural heritage, etc.

    In this paper we focus on satellite retrieval of aerosol properties, which requires that the particles are optically

active, i.e. their size is of the same order of magnitude as the wavelength of the incident light. The radiometers which are commonly used for aerosol measurements from space, making use of the Earth-reflected solar radiation at the top of the atmosphere (TOA), are not sensitive to particles smaller than about 100 nm due to their low scattering efficiency at wavelengths in the UV/VIS (Sundström et al., 2015). Very large particles, larger than some tens of μm, occur in very low concentrations and therefore contribute little to the radiance measured at TOA.

Hence the particles observed from satellites, during clear-sky conditions, are in the size range of ca. 100 nm to some tens of μm, and thus do not include newly formed particles (Sundström et al., 2015), unless proxies are used (Kulmala et al. 2011; Sundström et al., 2015). On the other hand, cloud droplets with sizes on the order of 10 μm do affect the TIR radiances, and this is used for cloud detection.

    China, with it's large variability of aerosol concentrations and a wide range of emissions of both different types

of aerosol particles and precursor gases as well as different climate regions and meteorological conditions, offers unique opportunities to study aerosols. Many studies have been published on aerosols in relation to air quality in the eastern part of China, including satellite remote sensing, ground-based measurements, modeling and combinations thereof, which often focus on local or regional aspects (e.g., Song et al., 2009; Wang, S., et al., 2011; Ma et al., 2016; Zou et al., 2017; Xue et al., 2017; Miao et al., 2017; Guo et al., 2017). Satellites offer the





opportunity to obtain information, using the same instruments and methods, over a large area during a longer period of time. In addition, using a lidar such as CALIOP/CALIPSO (Winker et al., 2007), complementary information can be obtained on the aerosol vertical structure (Winker et al., 2009) and hence a 3-D aerosol climatology can be developed (Winker et al., 2013). CALIOP also provides information on aerosol type (Omar et al., 2009).

The objective of the current study is to present the aerosol spatial and temporal distribution over mainland China, using two decades of satellite observations. The focus is on the use of the European Space Agency (ESA) Along Track Scanning Radiometers (ATSR), i.e. on ATSR-2 which flew on ERS-2 and provided data from 1995 until 2003, and the Advanced ATSR (AATSR) which flew on the environmental satellite ENVISAT from 2002 until April 2012 when contact to the satellite was lost and its mission ended. Hence, a period of 17 years of ATSR data is available. With the focus on aerosols over land, the AOD at a wavelength of 550 nm (AOD550, from here on referred to as AOD, unless specified otherwise) retrieved using the ATSR Dual View algorithm (ADV; Kolmonen et al., 2016; Sogacheva et al., 2017) was used. This data set is further extended to 2015 by using AOD data available from MODIS/Terra, Collection 6 (C6) (Levy et al., 2013). CALIOP data, available from 2007, was used to obtain information on the vertical distribution of the AOD (at 532 nm) over part of the study area. Thus a 3-D aerosol climatology over China was obtained, and the time dimension was added by evaluating a 2-decadal time series by combining ATSR and MODIS data spanning the period 1995-2015. The study encompasses the area between 18°-54° North and 73°-135° East, see Fig. 1, but the discussion focuses on AOD and vertical extinction profiles over China, i.e. the area over land within the Chinese border indicated by the blue line.

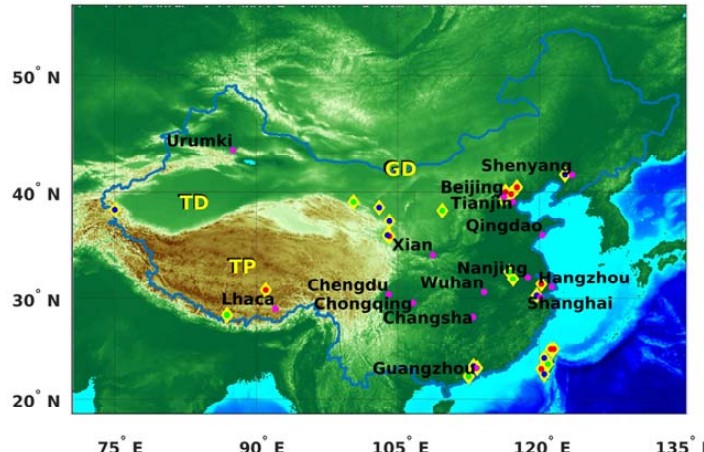

**Figure 1.** The study area is mainland China, i.e. the area within the Chinese border, indicated in this elevation map by the blue line. Also indicated are some major cities (purple dots) and the locations of the AERONET sites (yellow diamonds) used in this study for validation. The colour of the dot inside each diamond gives an indication for the length of the data record at that site (red: years, green: months, blue: days). Areas mentioned in this paper are the Taklimakan Desert (TD), Gobi Desert (GD) and Tibetan Plateau (TP). Other areas are the Beijing/Tianjin/Hebei (BTH) area, the Yangtze River Delta (YRD) including Shanghai and Nanjing, the Pearl River Delta in the south including Guangzhou and the North China Plain (NCP) including BTH, YRD and the area in between.

The ATSR dual view offers the opportunity to effectively separate the contributions of the surface and atmospheric reflections to the total reflection measured at TOA and thus retrieve aerosol properties independent of a surface correction, assuming that the ratio of the surface reflections in the forward and nadir views is independent of


wavelength (Veefkind et al. 1998; Kolmonen et al., 2016). However, that approach may fail over very bright surfaces such as snow, ice, or some desert areas. ATSR AOD data retrieved using the dual view algorithm ADV have been successfully applied in many studies (e.g., Veefkind et al., 1998; 1999; 2000; Robles-Gonzalez et al., 2000, 2003, 2006; 2008; Schmid et al, 2003; Sundström et al., 2012; Holzer-Popp et al., 2013; Virtanen et al.,

2014; de Leeuw et al., 2015; Rodriguez et al., 2015; Sogacheva et al., 2015; Popp et al., 2016; Sogacheva et al., 2017). By combining the ATSR-2 and AATSR data sets a unique time series of AOD over land, from June 1995 until April 2012, offers the opportunity to analyze the temporal variation of the AOD and possibly detect trends. However, the most recent changes in response to emission regulations in China (e.g., van der A et al., 2017) cannot be observed and the analysis of the ATSR time series remains inconclusive. Therefore, this time series has been

extended with MODIS/Terra C6 DTDB merged AOD data, selected because of the proximity over the overpass times of the ERS-2/ENVISAT and Terra satellites over China within ca. 1 hour. MODIS/Terra data is available from April 2000. Furthermore the most recent MODIS C6 data has been selected because of updates described by Levy et al. (2013). To use these data sets together requires an evaluation of their similarities and differences across the study area, where possible including an independent reference data set like the one provided by AERONET

(Holben et al., 1998) or CARSNET (Che, H., et al., 2015; Che, Y., et al.. 2016).

Earlier studies on the aerosol climatology and trends over China were made using ground-based remote sensing, i.e. sun photometers in CARSNET (Che, H., et al., 2015), hand-held sun photometers in the Chinese Sun Hazemeter network (CSHNET; Wang, Y., et al., 2011) and solar radiation measurements (e.g., Xu, X., et al., 2015), or satellite data, in particular MODIS (e.g., Li et al., 2003; Song et al., 2009; Wang, S., et al., 2011; Luo et

al., 2014; Tan et al., 2015; Xu, H., et al., 2015; Ma et al., 2016; He et al., 2017), but also using multiple satellite data (Lin et al., 2010; Guo et al., 2011; 2016; Dong et al., 2017; Zhao et al., 2017; Zhang et al., 2017). However, these studies used MODIS C5.1 AOD data and substantial differences exist between C6 and C5.1 (e.g., Levy et al., 2013; Sayer et al., 2014; Tao et al., 2015; Xiao et al., 2016). Furthermore, in addition to data sets over dark and brighter surfaces from the dark target (DT) and deep blue (DB) algorithms, respectively, C6 also provides a

merged DTDB data set based on criteria using the quality flags in each product (Levy et al., 2013). The merged DTDB data set offers better coverage but at the expense of a slight decrease in accuracy (Tao et al., 2015). Direct and systematic comparisons between C5.1 and C6 AOD over China have been published for the individual DT and DB data, but not for the merged DTDB AOD. This task is undertaken here to support the comparison of the results from the current study with those from previous work.

In this study the spatial distribution of AOD over China is presented and discussed as well as the vertical distribution of the aerosol extinction coefficients and AOD inferred from CALIOP data. Seasonal variations of AOD are presented and briefly discussed, as well as the 2-decadal time series (1995-2015) of AOD.

## 2. Aerosol data sets for China

Different data sources are used in this study, including satellite and ground-based observations. Satellite data sets

include AOD retrieved from ATSR and MODIS/Terra, and vertical profiles obtained from CALIOP. Independent and accurate ground-based data sets are used as reference for validation and evaluation. The study encompasses the area shown in Fig. 1, with focus on mainland China.



### 2.1 Satellite Data sets

#### 2.1.1 ATSR (ATSR-2 and AATSR)

The Along-Track Scanning Radiometer (ATSR) is a dual view instrument (near-nadir and 55° forward). The two views facilitate effective separation of surface and atmospheric contributions to the observed upwelling radiances, as applied over land in the ATSR dual view algorithm ADV (Kolmonen et al, 2016). Multiple wavelengths (7) from VIS to TIR facilitate effective cloud screening and allow for multi-wavelength retrieval of aerosol properties. ATSR has a conical scan mechanism with a swath of 512 km, resulting in daily global coverage in 5- 6 days. Data are provided with a nominal resolution of 1x1 km$^2$ sub-nadir and aerosol data are provided at a default spatial resolution of 10x10 km$^2$ on a sinusoidal grid (L2) and at 1° x 1° (L3).

ATSR-2 flew onboard ESA's ERS-2 from 1995-2003. The Advanced ATSR (AATSR) flew on ESA's environmental satellite ENVISAT and provided data from May 2002 until April 2012. Both satellites flew in a sun-synchronous descending orbit with a day-time equator crossing time of 10:30 LT (ERS-2) and 10:00 LT (ENVISAT). Together these instruments provided 17 years of global aerosol data. This time series is planned to be continued with a similar instrument on Sentinel-3, i.e. the Sea and Land Surface Temperature Radiometer (SLSTR), launched in the spring of 2016. A detailed description of the AATSR data processing, using the ATSR Aerosol Dual View algorithm ADV over land and the Single View algorithm (ASV) over ocean, is provided in Kolmonen et al. (2016).

The ATSR product used in this paper is the aerosol optical depth at a wavelength of 550 nm (hereafter referred to as AOD), over the study area for the full ATSR mission. The data were produced using ADV version 2.31 which includes cloud post processing as described in Sogacheva et al. (2017).

#### 2.1.2 MODIS

The MODIS (MODerate resolution Imaging Spectroradiometer) sensor (Salomonson et al., 1989) aboard NASA's Terra satellite has been flying in a near-polar sun-synchronous circular orbit for more than fifteen years (launched on 18 December 1999) observing the Earth-Atmosphere system. MODIS/Terra has a daytime equator crossing time at 10:30 LT (descending orbit), a viewing swath of 2330 km (cross track) and provides near-global coverage on a daily basis. One of the most successful products of MODIS, which has been used in numerous aerosol related studies, is the aerosol optical depth at 550 nm (hereafter referred to as AOD).

MODIS AOD is retrieved using two separate algorithms, Dark Target (DT) and Deep Blue (DB). In fact, two different DT algorithms are utilized, one for retrieval over land (vegetated and dark-soiled) surfaces (Kaufman et al., 1997; Remer et al., 2005; Levy et al., 2010, 2013) and one for retrieval over water surfaces (Tanré et al., 1997; Remer et al., 2005, Levy et al., 2013). The DB algorithm (Hsu et al., 2004, 2013) was traditionally used over bright surfaces where DT cannot be used (e.g. deserts, arid and semi-arid areas). However, the enhanced DB algorithm, which is used in C6, is capable of returning aerosol measurements over all land types (Sayer et al., 2013, 2014). The C6 DT expected error is $\pm(0.05+0.15\tau_{AERONET})$ over land and $+(0.04+0.1\tau_{AERONET})$, $-(0.02+0.1\tau_{AERONET})$ over sea relative to the AERONET optical thickness ($\tau_{AERONET}$) (Levy et al., 2013). The C6 DB expected error is $\sim\pm(0.03+0.2\tau_{MODIS})$ relative to the MODIS optical thickness ($\tau_{MODIS}$) (Hsu et al., 2013; Sayer et al., 2015).

Several changes have been made in C6 compared to C5.1. As the MODIS sensors suffer from degradation a new calibration approach has been used in order to remove major non-polarimetric calibration trends from the MODIS




data (Levy et al., 2013, 2015; Lyapustin et al., 2014). Details about these updates in C6 DT and DB data can be found in a number of recent studies (e.g. Levy et al., 2013; Tao et al., 2015; Sayer et al., 2015; Georgoulias et al., 2016). Another important update in C6 is the inclusion of a merged (DT and DB) dataset as described in Levy et al. (2013). This includes measurements from both algorithms, offers a better spatial coverage and can be used in quantitative scientific applications (Sayer et al., 2014). In this work, the merged C6 L3 MODIS/Terra (MOD08_M3) monthly 1° x 1° gridded AOD dataset is used for the period 3/2000-12/2015.

### 2.1.3 CALIOP

The LIght Detection And Ranging (lidar) is a powerful remote sensing technique for obtaining information related to the vertical distribution of aerosols in the atmosphere (Liu et al., 2002). On a global scale, lidar data are acquired by the Cloud Aerosol Lidar with Orthogonal Polarization (CALIOP) which is the primary instrument onboard the Cloud-Aerosol Lidar and Infrared Pathfinder Satellite Observations (CALIPSO) satellite (Winker et al., 2007). CALIPSO, developed as a collaboration project between NASA and the space agency of France (CNES), provides altitude-resolved profiles of aerosols and clouds since June 2006. In addition to the total attenuated backscatter signal at two wavelengths (532 nm and 1064 nm), CALIOP is capable of acquiring polarization measurements at 532 nm. Since the particle depolarization ratio is considered as the fingerprint of desert dust particles (Ansmann et al., 2003, Liu et al., 2008), CALIOP is an ideal instrument for studies related to the three-dimensional distribution and transport of dust in the atmosphere (Amiridis et al., 2013: Proestakis et al., 2017).

CALIPSO joined the A-Train constellation of satellites in April 2006 (Winker et al., 2007). Being an integral part of the A-Train formation, CALIPSO is in a sun-synchronous polar orbit, with a local equator crossing time at 13:30 and an orbit repetition frequency of approximately 16-days. On board CALIPSO the primary instrument is CALIOP, a dual-wavelength and dual-polarization elastic backscatter Nd:YAG lidar (Hunt et al., 2009). CALIOP transmits linearly polarized pulses at 532 nm and 1064 nm, while a telescope of 1 m diameter collects the backscatter signals. Based on the 532 nm and 1064 nm total backscatter signals and on the parallel and perpendicular polarization components of the 532 nm backscatter signal, CALIOP provides global and continuous information on the vertical distribution of aerosols and clouds (Winker et al., 2009). The product of CALIOP is provided in different levels of processing. Here, we use the L2 product, which provides height-resolved information of aerosol and cloud backscatter and linear depolarization ratio along the CALIPSO track. Based on a number of parameters, namely the magnitude of the attenuated backscatter, the ratio cross-to-total of the attenuated backscatter signals, the altitude of the detected layers and the surface characteristics along the CALIPSO orbit, the CALIPSO algorithm classifies the detected atmospheric feature types into subtypes (Omar et al., 2009). In case of aerosols the algorithm assigns aerosol-dependent lidar ratios (LR) to the different subtypes in order to convert the L2 backscatter coefficient profiles into profiles of extinction coefficient (Young and Vaughan, 2009).

### 2.2 Ground based reference data: AERONET

For the validation of satellite-retrieved aerosol products, AERONET sun photometer data (Holben et al., 1998) are most commonly used as an independent data source which are publicly available at the AERONET website



(http://aeronet.gsfc.nasa.gov/). An extensive description of the AERONET sites, procedures and data provided is available from this website. Ground-based sun photometers provide accurate measurements of AOD (uncertainty ~0.01–0.02, Eck et al., 1999) because they directly observe the attenuation of solar radiation without interference from land surface reflections. The parameter used in this study is the Version 2 L2.0 (cloud screened and quality assured) AOD at 550 nm, obtained from interpolation between AOD retrieved at 440 and 675 nm using the Ångström Exponent. The locations of the AERONET sites used in this study are indicated in Fig. 1 and their coordinates are listed in Table 1.

Table 1. AERONET sites used in this study

| AERONET name station | Latitude | Longitude |
|---|---|---|
| BackGarden_GZ | 23,3 | 113,0 |
| Beijing | 40,0 | 116,4 |
| Chao_Jou | 22,5 | 120,5 |
| Chen-Kung_Univ | 23,0 | 120,2 |
| EPA-NCU | 25,0 | 121,2 |
| Hangzhou-ZFU | 30,3 | 119,7 |
| Hefei | 31,9 | 117,2 |
| Jingtai | 37,3 | 104,1 |
| Kaiping | 22,3 | 112,5 |
| Lanzhou_City | 36,0 | 103,9 |
| Liangning | 41,5 | 122,7 |
| Lulin | 23,5 | 120,9 |
| Minqin | 38,6 | 103,0 |
| Muztagh_Ata | 38,4 | 75,0 |
| NAM_CO | 30,8 | 91,0 |
| NCU_Taiwan | 25,0 | 121,2 |
| PKU_PEK | 39,6 | 116,2 |
| QOMS_CAS | 28,4 | 86,9 |
| SACOL | 35,9 | 104,1 |
| Shouxian | 32,6 | 116,8 |
| Taichung | 24,1 | 120,5 |
| Taihu | 31,4 | 120,2 |
| Taipei_CWB | 25,0 | 121,5 |
| XiangHe | 39,8 | 117,0 |
| Xinglong | 40,4 | 117,6 |
| Yufa_PEK | 39,3 | 116,2 |
| Yulin | 38,3 | 109,7 |
| Zhangye | 39,1 | 100,3 |
| Zhongshan_Univ | 23,1 | 113,4 |



### 3. Data overview

#### 3.1.1 ATSR-retrieved AOD 1995-2012, using ADV v2.31

ATSR-2 retrieved AOD data are available for the period June 1995- December 2003, with some gaps in 1995 and 1996, while also toward the end the data were not reliable. Hence ATSR-2 data are used in this study only until

August 2002. AATSR data are available for the period May 2002 - April 2012, but some data are missing in 2002. The consistency between the ATSR-2 and AATSR data sets has been discussed in Popp et al. (2016). Years for which data are not continuously available for operational purposes are not shown in multi-year averaged maps, or aggregates. Here the term aggregate is used instead of average because of missing data for, e.g., cloudy situations, bright surfaces, and other situations where a successful retrieval was not achieved. Furthermore, satellite data are

biased to clear sky situations and hence no information is available for cloudy or partly cloudy scenes. In addition, satellite observations offer a snapshot during the overpass at a certain time of the day and, in the case of ATSR with a limited swath width, data are available only every 3-5 days, depending on latitude. Lacking information on the AOD for other days, the data cannot represent a true average.

A map showing the spatial distribution of the ATSR-retrieved AOD over China, aggregated for the full years

2000-2011, is shown in Fig. 2. This period was selected to allow for comparison with MODIS/Terra. Differences with the aggregated AOD map for the ATSR full-mission period, 1995-2012, are very small (not shown). It is clear that in such aggregate the absolute AOD value may not be representative for the actual value in a certain area because systematic temporal and year-to-year variations are hidden in the process. Temporal variations will be discussed below based on time series.

The map in Fig. 2 shows the commonly reported high AOD over SE and SW China with the highest values (on the order of 0.8) over the North China Plain (NCP) and the Sichuan province. Also south of the Himalayas the AOD is high, with moderately high AOD over the area east of the Himalayas and SW China. NW of the NCP the AOD is moderate with values around 0.3. In most other areas the AOD is low with values of 0.15 and smaller. In the west, over the Taklimakan desert, the multi-year aggregated AOD is high, due to the presence of wind-blown

desert dust in the spring. However, the highest values are not observed over the bright desert surface where the ADV retrieval was not achieved and thus no data are available (white area). Along the Chinese coast the AOD is high (on the order of 0.5), with overall smooth land-sea transitions, and decreasing toward open ocean. Likely the high AOD is due to a combination of transport from land and ship emissions along this very busy shipping route. A very high AOD area is observed at ca. 34° N, 122° E which also occurs in the MODIS data (see below) as well

as in OMI-retrieved $NO_2$ column data (Ding et al., 2017). Likely, ship emissions are also the reason for the AOD hotspot at 38°N, 119° E where $NO_2$ concentrations are also high.

It must be kept in mind that these are 12 year-aggregated values and strong deviations may occur in certain years or seasons.



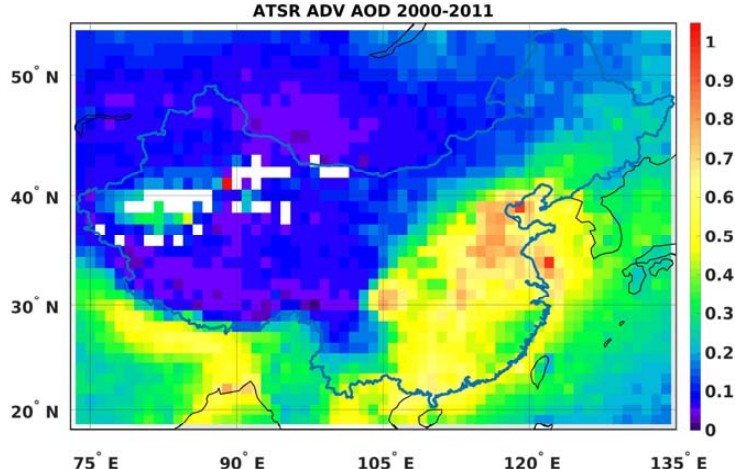

**Figure 2. Spatial distribution of the AOD over China, aggregated over the (full) years 2000-2011. The AOD has been retrieved from ATSR-2 and AATSR data, using the ADV algorithm v2.31. The AOD scale is presented in the colour bar to the right of the map. Areas for which no data are available are shown in white.**

**3.1.2 MODIS/Terra 2000-2015: C6 merged DTDB AOD**

The MODIS AOD used in this study is the MODIS/Terra C6 L3 DTDB merged AOD product for the period April 2000 until December 2015. Here MODIS/Terra has been chosen because of the morning orbit with an equator overpass (descending) time at 10:30, i.e. close to the ATSR equator overpass times (ATSR-2 at 10:30 LT; AATSR at 10:00 LT), which allows for comparison of the data over China within about 1 hour. It is noted that the drift in

the MODIS/Terra blue channel has been corrected from C5.1 to C6 (Levy et al. 2013).

The spatial distribution of the MODIS/Terra DTDB AOD for the period 2000-2011 is presented in Fig. 3. The overall AOD distribution is similar to that presented in Fig. 2 for ATSR, but with some noticeable differences. The main features in Fig. 3 are the higher AOD provided by MODIS, as compared to ATSR, over almost the whole study area, as well as the much higher AOD over most of the Tibetan Plateau and the Taklimakan desert.

Obvious, the latter is due to the use of the DB algorithm over the bright areas in west China where DT provides hardly any AOD data (cf. Tao et al., 2015) but also the ADV-retrieved AOD is very low (over the Tibetan Plateau) or not available due the bright surface. The AOD spatial pattern over SE and North China is similar for MODIS and ATSR, with MODIS AOD higher, and the same is observed over Northern India. In contrast, the AOD retrieved by MODIS over Vietnam and Laos is lower than that retrieved using ATSR data. The differences

between ATSR and MODIS will be further addressed in Sect. 4.



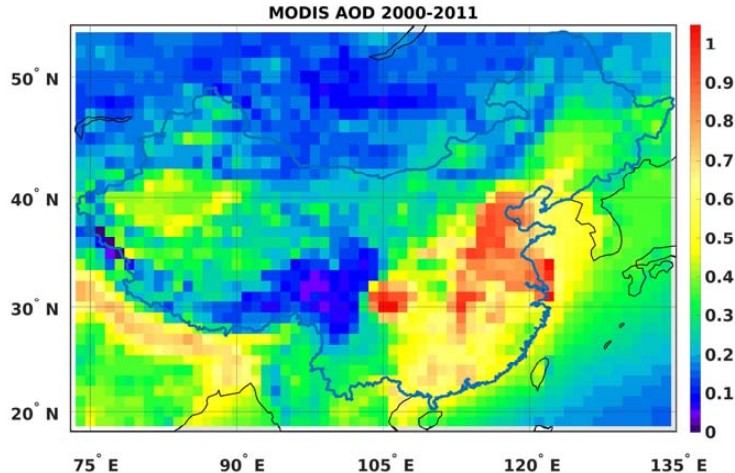

**Figure 3.** Spatial distribution of the MODIS/Terra C6 DTDB merged AOD over China, aggregated for the full years 2000-2011.

**MODIS C5.1 vs C6**. Many studies on the AOD over China have been published, as mentioned in the introduction.
These studies are relevant for comparison with the current study with regard to both the spatial resolution and the temporal behaviour. In view of the rather recent production of the MODIS C6 data, most of the earlier studies used C5.1. Levy et al. (2013) made an initial comparison between C6 and C5.1 for 4 months of MODIS/Aqua AOD data showing that the C6-C5.1 AOD difference is smaller than 0.1. Levy et al. also compared MODIS/Terra aggregated 1° x 1° AOD for one month (July 2008) and noted an extra C6-C5 difference over land in the
MODIS/Terra data. Their Fig. 22 shows that over China the Terra C6-C5 difference is larger than 0.1.
Sayer et al. (2014) did not specifically address the AOD over China although these authors noted that in C6 DB AOD tends to be lower than DT in the high-AOD region of China throughout the year. They also concluded that the merged product does not specifically outperform the DT or DB results. The Terra C6 3 km AOD product was validated by Xiao et al. (2016) using AERONET data from the DRAGON Asia campaign (2012-2013) (Holben
et al., 2017) and over Beijing using hand-held sun photometers.
Tao et al. (2015) evaluated MODIS/AQUA C6 AOD over different regions in China for both DB and DT products, but not for the merged DTDB AOD, using AERONET data as reference. One handicap is the sparsity of AERONET stations and their spatial distribution across China and another one is the length of operation of each station. Tao et al. considered five different regions, which are indicated as northern China, the Yangtze River
Delta (YRD), southern China, north western China and scattered arid areas. It is noted that most AERONET stations were concentrated in the northern China and YRD regions. The validation results show the good performance of DB for the northern sites, whereas over all other sites DB underestimates the AOD. DT overestimates the AOD over almost all AERONET sites where sufficient quality retrievals are available to allow for proper evaluation. Seasonal mean DT AOD over eastern China may be 0.3-0.4 higher than DB. Tao et al. also
note that DT misses haze periods with high AOD and further comment on the use of NDVI in the merging procedure.
In view of the regional differences in the behaviour of DB and DT, the regional quality of the merged product cannot be assessed a priori and depends on whether the DB or DT product has been selected, and when both are



used, on their actual values in the merging process. The merged DTDB AOD from both Terra and Aqua were evaluated by Zhang et al. (2016) for two CSHNET (Xin et al., 2007) sites near Beijing, i.e. Beijing City and a rural mountain site west of Beijing (Beijing Forest). The results show the good performance of the merged DTDB AOD product although it is not better than any of the individual DT or DB products in all cases. At the rural site,

DTDB performs similar to DT but better than DB, as compared to the CSHNET AOD, with a slight underestimation of the daily product over the rural site. Over the urban site the DB product performs somewhat better than DTDB, with a slight overestimation of the latter.

In this study, a C5.1 DTDB merged AOD product has been produced over China following the procedure described in Levy et al. (2013). The difference between the C6 and C5.1 merged products is presented in Fig. 4.

Figure 4 clearly shows the higher C6 AOD over most of eastern China as well as the Tibetan Plateau with the largest differences, up to 0.2, over the NE of the NCP and Sichuan province and the western part of the Tibetan plateau. On the other hand, C6 is lower over the lower part of western China and in particular the Taklimakan desert where local differences are observed of -0.25. These differences will be further discussed after evaluation of the satellite AOD products versus available AERONET reference AOD data.

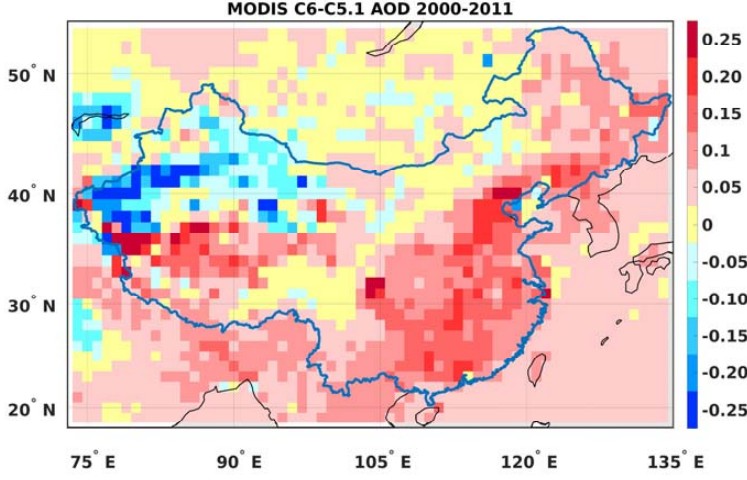

Figure 4. Difference between MODIS C6 and C5.1 merged DTDB AOD, see colour scale at the right. Red colours indicate that C6 is higher, blue colours indicate that C5.1 is higher.

### 3.1.3 CALIOP 2007-2015: The three-dimensional distribution of aerosols

In addition to the presented and discussed horizontal variability of ATSR and MODIS columnar properties,
CALIOP observations are synergistically used in this study, in order to provide information on the vertical distribution of aerosols over China. The vertical distribution of aerosols in the atmosphere greatly affects aerosol-cloud interaction (DeMott et al., 2009; Hatch et al., 2008), is critical to estimations of the aerosol direct and indirect radiative forcing on climate (Haywood and Boucher, 2000), to human health and degradation of air quality (Goudie, 2014).

The horizontal variability of the CALIOP-derived AOD at 532 nm is shown in Fig. 5 for the domain 35º-45º N and 70 º-150º E, for winter (DJF), spring (MAM), summer (JJA) and autumn (SON), where the data for each season has been averaged over the years 2007-2015. The vertical distribution of aerosols over the same domain


(35°-45° N, 55°-155° E) and for the same seasons is presented through the climatological extinction coefficient profiles at 532 nm (Fig. 6). Through the combination of the vertical dimension (Fig. 6) with the horizontal AOD distribution (Fig. 5) the full 3-D overview of atmospheric aerosols over this domain is provided. Similar figures for other areas, encompassing SE Asia (5°-55° N, 65°-155° E) are provided in Proestakis et al. (2017).

The domain shown in Fig. 5 encompasses the Taklimakan and Gobi Deserts and the densely populated Beijing-Tianjin-Hebei (BTH) area. Over this domain, similar patterns of AOD are observed throughout the year, with their intensity varying strongly with the seasons, especially over the dust sources. The Taklimakan/Gobi Deserts and the BTH area are clearly mapped through the high AOD values. Large mean AOD values, of the order of 0.3-0.8 are observed over the arid region of the Taklimakan Desert/Tarim Basin and the BTH area, while the semi-arid

Gobi Desert yields significantly lower mean AOD values, of the order of 0.1-0.3. Over the Taklimakan Desert the highest AOD values, of the order of 0.5-0.8, are observed during MAM and JJA, while during the period between September and February AOD is much lower (~0.3). The seasonal variation over the Gobi desert is similar to that over the Taklimakan Desert, but AOD values are substantially lower, even at its maximum activity (<0.3). The observed seasonality and variability of the AOD over the Taklimakan/Gobi Deserts and the high AOD values

observed during MAM over the Taklimakan Desert are strongly related to the activation mechanisms of the dust sources (Prospero et al. 2002), the local topography of the Tarim Basin (Yumimoto et al., 2009) and the cyclonic systems developed over Mongolia (Sun et al., 2001). Downwind from the Taklimakan and Gobi dust sources the anthropogenic activity in the densely populated and highly industrialized BTH area results in a consistently high average AOD which is present throughout the year. Similar AOD features are observed between all four seasons

with larger AOD values, of the order of 0.7, during JJA and lower AOD values, of the order of 0.5, during SON.

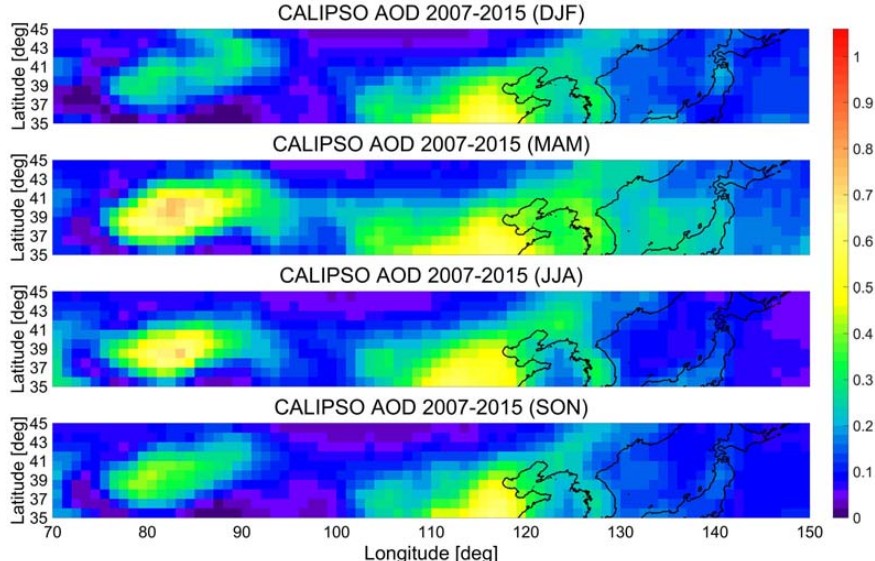

**Figure 5. Seasonal maps of AOD at 532 nm derived from CALIOP over the area 35º-45º N, 70º-150º E, including the Taklimakan/Gobi deserts and the Beijing-Tianjin-Hebei area, derived from 9 years of CALIPSO overpasses (2007-2015): winter (DJF), spring (MAM), summer (JJA) and autumn (SON).**

The CALIOP-derived 9-years averaged vertical distribution of the climatological extinction coefficients over the same domain as discussed above is shown in Fig. 6, for each of the four seasons in the period 2007-2015. Over


the Taklimakan Desert and the BTH area, similar climatological extinction coefficient features are observed close to the surface, with values as high as 200 Mm$^{-1}$ persistently present throughout the year. Over the vast semi-arid Gobi Desert though, the near-surface climatological extinction coefficient values are significantly lower (~50 Mm$^{-1}$). The extinction coefficient values are altitude dependent and distinctly decrease with height. Over BTH

and eastwards (110°-130° E) high extinction coefficient values (~200 Mm$^{-1}$) are in general suppressed below 1 km above sea level (asl), while over the Taklimakan Desert (77°-86° E) high extinction coefficient values of the order of 200 Mm$^{-1}$ are observed as high as 3 km asl. The eastward long-range transport of dust aerosols generated from the Taklimakan and Gobi Deserts is also evident. At altitudes higher than 3 km asl over the BTH area high dust-related extinction coefficient values (~125 Mm$^{-1}$, gradually decreasing with height) are observed over the

BTH area. This suggestion is supported by the areas of high extinction values (Taklimakan, ca. 200 Mm$^{-1}$; Gobi, ca. 75 Mm$^{-1}$) located to the west of BTH. Dust extinction coefficient profiles (Proestakis et al., 2017) suggest that part of the transported aerosol is dust, with a large contribution of locally produced aerosols over BTH. Over the whole region the extinction decreases gradually with height above the surface up to an altitude of ca. 8 km, where the extinction coefficient has decreased to ~10 Mm$^{-1}$. However, the height of the layer gradually decreases toward

the east, following the surface elevation to some degree, but also with gradually increasing layer depth. The synergy of the horizontal (Fig. 5) and vertical (Fig. 6) distributions allows for the simultaneous study of the emission sources (Taklimakan/Gobi Deserts, BTH area), the aerosol load and the corresponding injection height of dust aerosols in the atmosphere.

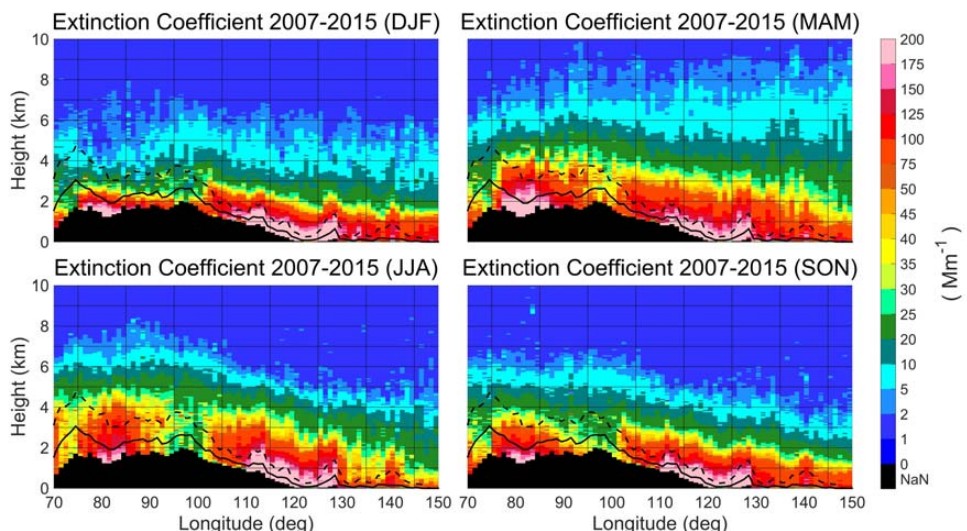

**Figure 6. Vertical distribution of the climatological extinction coefficient profiles at 532 nm over the area 35º-45º N, 70º-150º E, including the Taklimakan desert and the Beijing-Tianjin-Hebei area, derived from 9 years of CALIOP measurements (2007-2015): winter (DJF), spring (MAM), summer (JJA) and autumn (SON). The solid and broken lines indicate the mean and maximum elevation of the aerosol extinction above the surface.**

## 4. Validation and evaluation

The data overview presented above shows the differences between the AOD derived from ATSR, MODIS and CALIOP data. To evaluate the quality of the ATSR and MODIS retrieved AOD, these products are compared



with reference AOD data available from AERONET sites in the study area (see Fig. 1 and Table 1 for locations). For this comparison collocated data are used, i.e. satellite data within a circle with a radius of 0.125º around the AERONET site are averaged and compared with averaged AERONET data measured within ± 1 hour of the satellite overpass time.

**4.1 ATSR**

AATSR AOD data sets over China produced by three different algorithms, including ADV,  have been validated by Che, Y., et al. (2016) using as reference the AOD data from selected AERONET and CARSNET (Che et al., 2015) sites for the years 2007, 2008 and 2010. The results show that the AATSR-retrieved AOD is underestimated by a factor which increases with increasing AOD. However, Che et al. likely used an older version of the ADV-

retrieved data set than the one produced by v2.31 which is used in the current work. The v2.31 AOD is substantially different from the earlier version as shown in Sogacheva et al. (2017), especially for high AOD regions. (Che et al. do not mention which ADV version was used, but at their submission date the latest updates described in Sogacheva et al. (2017) had not been implemented). Sogacheva et al. present a validation of the full-mission AATSR AOD over China versus AERONET AOD. Fig. 7 shows a density scatterplot of the ATSR-

retrieved AOD versus AERONET AOD for the sites listed in Table 1. Comparison of the AOD scatterplots in Che et al. (their Fig. 2a) with those in Fig. 7 shows the better performance of the newer ADV version v2.31 over China although a small underestimation of less than 0.1 remains for AOD up to ~0.5, increasing somewhat with increasing AOD between 0.5 and 0.9. For AOD larger than 1.4, ADV v2.31 overestimates with respect to the AERONET reference AOD and in view of the low number of valid data points and their large scatter the use of

these data is not recommended.



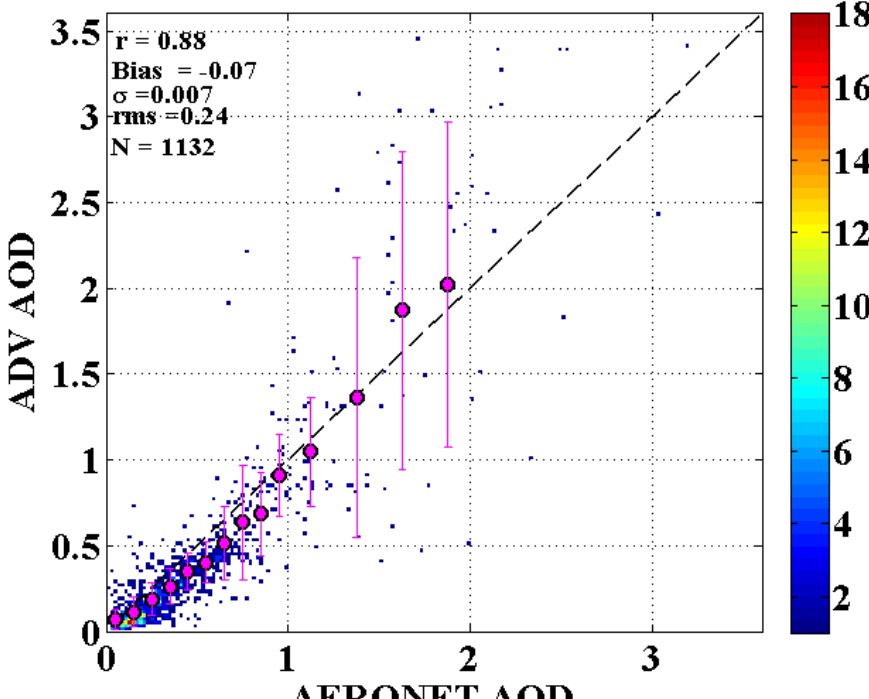

**Figure 7. Density scatterplot of ATSR-retrieved AOD, using ADV v2.31, over China for the years 2002-2012, versus AOD from AERONET stations in mainland China (cf. Fig. 1 and Table 1). The filled circles are the averaged ATSR AOD binned in 0.1 AERONET AOD intervals (0.2 for AERONET AOD>1.0) and the vertical lines on each circle represent the 1-sigma standard deviation of the fits. Statistics in the upper left corner indicate correlation coefficient r, bias, standard deviation, root mean square (rms) error and number of data points (N). The colour bar on the right indicates the number of data points.**

### 4.2 MODIS/Terra C6 merged DTDB AOD

MODIS AOD over China has been validated versus AERONET and CARSNET AOD. However, as discussed in Sect. 3.1.2, only few publications address the validation of the MODIS C6 data suite and most of these consider only the AQUA DT and DB data set. The validation of the MODIS/Terra C6 merged DTDB L2 AOD product is shown in the density scatterplot of Fig. 8, where MODIS/Terra AOD has been plotted vs AERONET AOD using the available AERONET sites listed in Table 1. The MODIS AOD has been binned in AERONET bins with a width of 0.1 showing the good agreement between MODIS and AERONET data for AOD up to 1.8 but with a slight overestimation on the order of 0.1, for AOD up to 0.5 and somewhat more for higher AOD.



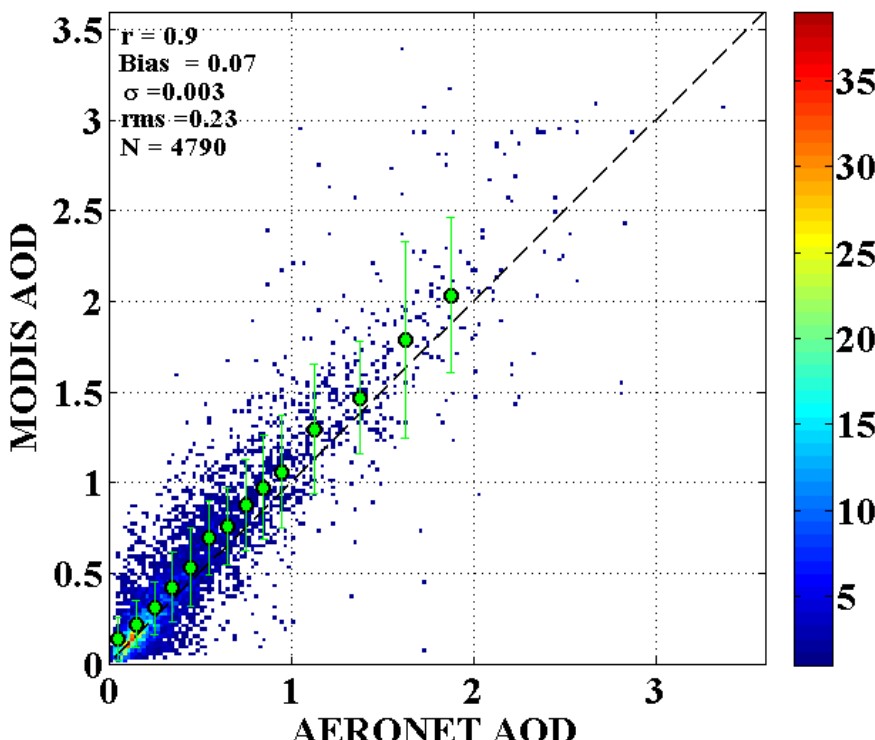

**Figure 8: Same as Fig. 7, but for MODIS/Terra C6, merged DTDB AOD data.**

**4.3 Intercomparison of AATSR and MODIS/Terra C6 DTDB merged AOD**

Having established that both the ADV and the MODIS/Terra C6 DTDB merged AOD data sets compare well with

5    the AERONET reference AOD data, we can address the differences observed in the AOD maps in Figs. 2 and 3, with MODIS AOD overall higher than that from ATSR. The difference map in Fig. 9 shows the actual differences between the two data sets (ATSR-MODIS) aggregated over the overlapping years (2000-2011) which are largest over brighter areas, such as the Taklimakan Desert and the Tibetan Plateau, where MODIS DTDB is governed by the DB data, which underestimates the AOD with respect to AERONET (Tao et al., 2015; 2017) whereas ADV

10    and especially DT provide few successful retrievals. However, also over areas with very high AOD, such as Sichuan province, NCP and YRD, the differences are large. In contrast, along the mountains from the NW to the SW of China and around Sichuan province, the AATSR and MODIS AOD are in very good agreement, within ±0.05, i.e. the estimated retrieval uncertainty over land (for MODIS). In other areas the AOD difference is ca. 0.1, as may be expected from the validation presented above, i.e., showing that overall MODIS slightly overestimates

15    and AATSR slightly underestimates with respect to AERONET which adds up to the AOD difference of ca. 0.1.



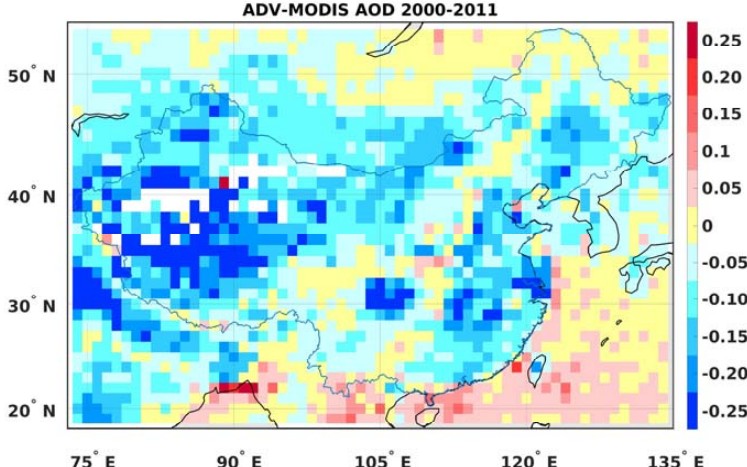

**Figure 9. Difference of ATSR AOD minus MODIS/Terra C6 DTDB merged AOD, aggregated for the years 2000-2011, over China. The values in increments of 0.05 are given in the colour scale to the right.**

The surprising finding is thus the high AOD difference over SE China, i.e. mostly over the low elevation part of
China (Liu et al., 2003) classified as forests and cropland (Bai et al., 2014), i.e. dark surfaces where retrieval
algorithms are expected to perform best. This is also the area where most of the AERONET Sun Photometers are
located, with clusters in the BTH area and the YRD. Apparently, the slight under- and over-estimations by ATSR
and MODIS, respectively, are not due to the surface correction in the retrieval algorithms and likely caused by
the aerosol types used.

Figure 10 shows a scatterplot of MODIS AOD vs AATSR AOD, for collocated AATSR-MODIS-AERONET
data, i.e. only AOD data are shown for AERONET sites where both MODIS and AATSR have achieved a
successful retrieval and the overpasses were within ± 1 hour while also AERONET data were available, for the
period 2002-2012. The colour code indicates the difference in the exact overpass time. Figure 10 shows that the
difference in exact overpass time, varying between 20 and 80 minutes, does not lead to a systematic effect on the
MODIS/AATSR AOD differences. Hence the somewhat later MODIS/Terra overpass, which could influence the
AOD as a result of a developing atmospheric boundary layer during the morning hours and associated temperature
and relative humidity difference, does not explain the higher MODIS AOD.



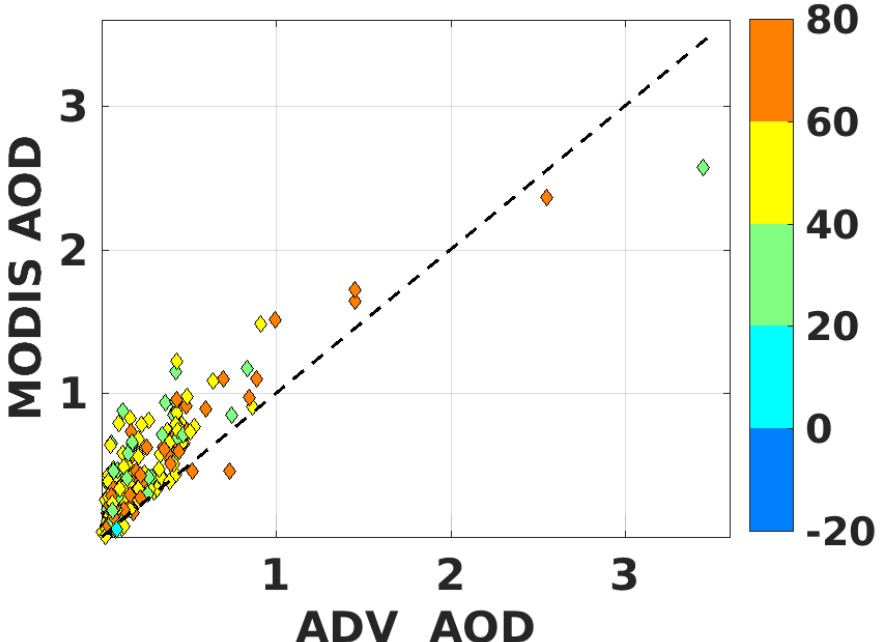

**Figure 10. MODIS/Terra C6 DTDB merged AOD versus AATSR ADV v2.31 AOD, for collocated AATSR-MODIS-AERONET data, as described in the text of Sect. 4.3. The colours, see scale at the right, indicate the difference between the MODIS/Terra and ATSR overpass times in minutes.**

## 4.4 CALIOP

The comparison of the CALIOP data over the region 35º-45º N over China with ATSR and MODIS data in Figs. 2 and 3, or better in Fig. 11 which shows the seasonal variations of the AOD retrieved from ATSR and MODIS data, shows similar patterns with high AOD over the Taklimakan Desert and the BTH area (differences between ATSR and MODIS were discussed in Sect. 4.3). However, differences are also observed, i.e. the underestimation of CALIOP AOD compared to AOD retrieved by MODIS/Terra. As mentioned above, the CALIOP AODs refer to the years 2007-2015, while ATSR and MODIS/Terra AODs refer to the years 2000-2011. Hence a direct comparison of the spatial distributions of the AOD should not be made since substantial year-to-year variations may occur, depending on meteorological and synoptic conditions. Furthermore, CALIPSO overpasses in the afternoon whereas ENVISAT and TERRA overpasses in the morning. The differences between the ATSR, MODIS/Terra and CALIOP AOD are likely due to the highly non-uniform data sample and to the fundamentally different algorithms and operation of the sensors. Following the literature, the CALIOP aerosol extinction coefficients are slightly underestimated as compared with EARLINET lidars (Papagiannopoulos et al., 2016). Tian et al. (2017) obtained similar results from comparison with lidar measurements in SACOL (China). A comparison of the CALIOP AOD climatological product against spatially and temporally co-located AERONET observations is discussed in Amiridis et al. (2015). In their Fig. 15, the absolute bias of the means between the CALIOP optimized product (named LIVAS) and AERONET reveals biases within ±0.1 in terms of AOD.



## 5. Seasonal variation

The spatial variation presented in the previous chapter is strongly influenced by emissions and meteorological factors which obviously vary seasonally due to both natural processes and human activities. As a result, strong seasonal variations are observed in the AOD distributions as shown in Fig. 11 where the ATSR and MODIS-retrieved AOD over the study area are shown for winter (DJF), spring (MAM), summer (JJA) and autumn (SON). As mentioned above, ATSR retrieval is often not successful over bright surfaces and has hardly any coverage over the Taklimakan and Gobi deserts and the Tibetan Plateau and also in the North during the winter. Due to the use of the DTDB merged AOD, MODIS has better coverage over bright surfaces and thus higher AOD over the Tibetan Plateau and the Taklimakan and Gobi deserts as well as in the north of China during the winter season. Seasonal AOD maps from CALIOP for the region 35°-45° N over China were shown in Fig. 5 and the vertical climatological extinction coefficient profiles for the same region were presented in Fig. 6. CALIOP, ATSR and MODIS/Terra are used synergistically for the analysis of the regional and seasonal variation of the AOD over China. The horizontal variability of CALIOP AOD (Fig. 5) shows similar features to the AOD distributions provided by ATSR and MODIS/Terra (Fig. 11), i.e. high AOD values over the Taklimakan Desert and over the BTH area, with lower values over the Gobi Desert. However, differences also exist as discussed in Sect. 3.1.3.

The AOD spatial distribution over China differs between seasons. The highest AOD is observed in the summer time over the NCP including the BTH area, with AOD of the order of 0.9, and somewhat lower values (ca. 0.6) in the spring and autumn and minima in the winter. This seasonal behaviour is in contrast to that of near-surface aerosol concentrations, indicated for instance by PM2.5, i.e. the mass concentration of dry particles with in situ diameters smaller than 2.5 μm, which peak in the winter and reach a minimum in the summer (e.g., Wang et al., 2015). These differences can be explained by seasonal variations in synoptic patterns (Xia et al., 2007; Miao et al., 2017), and associated boundary layer height and relative humidity, transport of aerosols, emission of primary aerosols and aerosol precursors contributing to chemical processes leading to secondary formation of new aerosol particles and thus higher concentrations (e.g., Tang et al., 2016). In particular the seasonal emissions of dust aerosol (spring) and biomass burning aerosol (season varies with region) have a strong influence. Song et al. (2009) also noted that the temporal correlation between the monthly values of the PM10 concentrations and MODIS AOD exhibit regional seasonality contrasts over China, with correlation coefficients > 0.6 near the southeast coast and -0.6 or lower in the north-central regions. They attribute this contrast to differences in the aerosol size distributions and support their argument by MODIS data on the distribution of the Ångström exponent and fine mode fraction.

Relatively high AOD is observed over the Sichuan province and Chongqing in all seasons, with values of ca. 0.8 and somewhat lower in the autumn. Also over the Pearl River Delta (PRD) the AOD is relatively high throughout the year with values on the order of 0.4-0.6 in winter, summer and autumn and a maximum in the spring of ca. 0.8. In the Shanghai/Nanjing region in the Yangtze River Delta (YRD) and the area to the SW of YRD toward PRD, the AOD is relatively high in the spring, summer and autumn, with values of ca. 0.5-0.6 and lower in the winter (around 0.3). South of the YRD, along the coast, an area is observed with more moderate AOD of about 0.3 throughout the whole year, which however stretches inland further in the winter then in the summer and autumn and is smallest in the spring. Over sea, along the SE coast of China, the AOD is on the order of 0.5 in the area north of the YRD, while further south it varies between about 0.3 and 0.5. Also the extent of the elevated AOD area varies with the season and is largest in the spring when it reaches east beyond Taiwan. It is noted that





the BTH, PRD and YRD regions host three major urban clusters that constitute huge spatial sources of anthropogenic aerosols (Kourtidis et al., 2015).

This brief discussion clearly illustrates the seasonal variation of the AOD, as well as the regional differences in the seasonal variation. These will be further discussed in a separate paper (Sogacheva et al., in preparation).

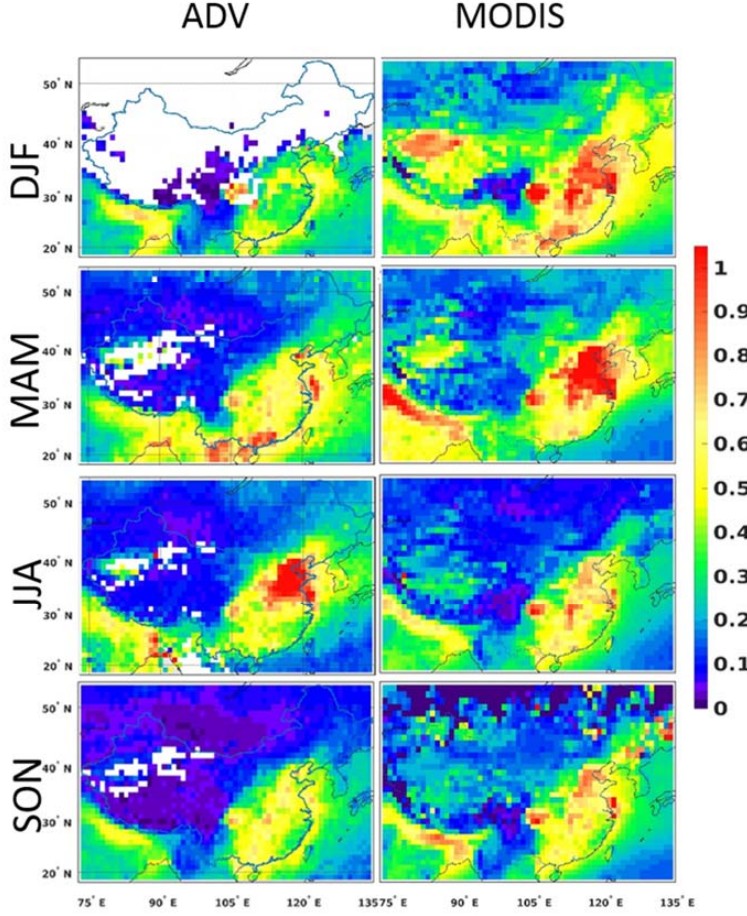

**Figure 11. Seasonally averaged maps of the ATSR (left) and MODIS (right) retrieved AOD distribution over China for the years 2000-2011: winter (DJF), spring (MAM), summer (JJA) and autumn (SON). The AOD colour scale is on the right.**

**6. Decadal time series: ATSR & MODIS (1995-2015)**

10    The combined ATSR-2 and AATSR AOD data set provides a continuous record of 17 years from 1995 until 2012. The ATSR time series, for yearly averaged AOD over mainland China as defined in Fig. 1, is presented in Fig. 12. The first two data points, for 1995 and 1996, miss the winter and spring months and may therefore be not fully representative. This time series shows a strong initial increase in the AOD by 50 %, from about 0.18 in 1995 to 0.27-0.28 in 2001 and 2003, but with a dip in 2002. It is noted that 2002 is the transition between ATSR-2 and



AATSR and the data point presented in Fig. 12 is the average of 7 months (Jan-July) of ATSR-2 AOD and 5 months (Aug.-Dec.) of AATSR AOD.

This initial increase is followed by a strong decrease in 2004 after which the AOD seems to increase to a plateau of about 0.27 in 2006-2008. In 2009 and 2010 the AOD is lower by about 0.03-0.04 and then increases again to

ca. 0.27 in 2011. The interruption of the ATSR time series after 2011 is unfortunate because this year could be a tipping point where the AOD starts to decrease, as shown by Zhao et al. (2017) for East and Central China. Therefore, to visualize the evolution of AOD over all China beyond the ATSR era, the MODIS/Terra time series has been added in Fig. 12. It is noted that the first MODIS data point for 2000 may not be fully representative since the first three months are missing.

Clearly, as described above, there are differences between ATSR and MODIS/Terra AOD and the MODIS/Terra yearly averaged AOD is 0.1-0.2 higher than that of ATSR. However, as Fig. 12 shows, the two curves show similar features with minima and maxima in the same years. The offset between the two data sets is similar for most of the period 2003-2011. Hence, in spite of this offset, the temporal behaviour of the AOD is well represented by both data sets, for the overlapping period. The clear advantage of using both data sets is that ATSR provides

additional information on the pre-EOS period, showing the initial increase of the AOD before 2000. MODIS/Terra complements the time series for the post-ENVISAT period, showing a decrease after 2011.

The observed AOD variations may be in response to the enforcement of policies to reduce emissions, but they may also be caused by meteorological influences, or a combination of these and other factors. The initial growth between 1995 and 2001 reflects urbanization and economic growth (e.g. Hao and Wang, 2005) and was also

reported by Guo et al. (2011) using TOMS-retrieved AOD and by Hsu et al. (2012) using SeaWiFS. The initial decline after 2003 may be due to emission control measures such as those implemented in Beijing and reducing atmospheric concentrations of PM10, $SO_2$ and $NO_2$ (Hao and Wang, 2005). The AOD minimum in 2002 is observed in both the ATSR and MODIS/Terra data in Fig. 12, but the MISR AOD over the NCP presented by Liu et al. (2013) shows a clear maximum. Liu et al. relate the variability in the AOD over the NCP to large scale

periodic climate variability modulated by the El Niño Southern Oscillation (ENSO) with a period of 3-4 years. Several authors identify a pivot point around 2006-2008 (Kang et al., 2016; Zhang et al., 2017; Zhao et al., 2017) after which the AOD fluctuates before the decrease sets in from 2011 (Zhao et al., 2017). Using a different analysis method, Zhang et al. (2017) suggest a decrease starting from ca. 2006-2008. It is noted that these studies were made for east or SE China with much higher AOD, as shown in Figs. 2 and 3, than that in Fig. 12 which are

averages over all China. The data in Fig. 12 show a decrease in the MODIS data for the period 2006-2009, as observed by Zhang et al. (2017), but the ATSR AOD does not show a clear decrease in that period. The decrease in both data sets in 2008-2009 could be the result of the economic recession as suggested by, e.g., Lin et al. (2010), He et al. (2016) and Zhao et al. (2017). The data in Fig. 12 do however suggest the onset of a decrease in 2011, confirming the conclusion by Zhao et al. (2017). This behaviour with pivot points in 2006 and 2011 is in line with

the reduction of emissions of aerosol precursor gases, such as $SO_2$ and $NO_2$ (van der A et al., 2017), possibly together with large scale climate variability as discussed above. The increase/decrease of anthropogenic particles is expected to increase/decrease the water uptake from the aerosols and hence the recorded AODs (Pozzer et al., 2015).



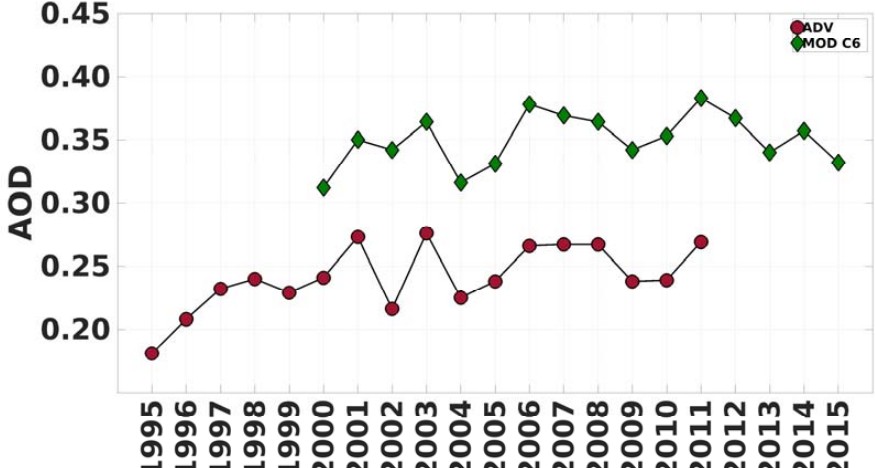

**Figure 12.** Time series of ATSR and MODIS retrieved AOD over China for two decades: 1995-2015. Note that data are missing in the beginning of the ATSR-2 observation period in 1995, and also MODIS/Terra data are available starting in April 2000; AATSR data start in May 2002. Therefore, the ATSR-2 1995 data point includes data for June-December and for 1996 July-December are included. ATSR-2 was used for 1995-7/2002, for 8/2002-2011 AATSR data were used. AATSR for 2012 is not shown since only 3 months of data are available, mainly in the winter (JFM) with low coverage. The MODIS 2000 data point includes data from April to December.

## 7. Discussion

Satellite data have been used to provide a 3-D aerosol climatology over mainland China for two decades (1995-2015), describing the spatial variation of the column-integrated extinction, or AOD, by combining ATSR-2, AATSR (both using the most-recent data set produced by the ADV v2.31 algorithm) and MODIS/Terra C6 DTDB merged AOD data. The vertical dimension is provided by the CALIOP extinction measurements since 2007 and the temporal variation has been provided by the time series of the yearly AOD. Inspection of the AOD data from different sources shows strong differences due to the failure of retrieval algorithms to deal with very bright surfaces, except for the MODIS DB algorithm which was designed for this purpose. However, DB and DT are complementary because DT performs better than DB over dark surfaces and the merged DTDB provides better coverage. Although the ATSR dual view algorithm was designed to eliminate the effect of surface reflectance on the radiances measured at TOA, it does not work well over very bright surfaces (Flowerdew and Haigh, 1995; Kolmonen et al., 2016). The differences between the ATSR and MODIS/Terra retrieved AODs have been addressed in detail by validation and evaluation of the individual AOD data sets versus a reference data set provided by AERONET. This study presents the first extensive validation of MODIS/Terra C6 DTDB merged and ATSR v2.31 AOD data over China. The results show that both data sets are of high quality. ATSR slightly underestimates and MODIS/Terra C6 slightly overestimates AOD, with respect to AERONET AOD, resulting in an overall higher MODIS than ATSR AOD and the difference increases as AOD is larger. We have no explanation for this behaviour and a more detailed study on the differences between the MODIS and ATSR algorithms is beyond the scope of the present study. Likely they are due to the choice of the aerosol models and/or cloud screening, while also calibration issues may influence the results. The data in Fig. 10 are collocated MODIS/Terra, ATSR and AERONET data, and it is unlikely that the results from these instruments together, with different cloud





screening criteria, are cloud-contaminated. MODIS, with its wider swath than ATSR, provides a much larger data sample and has global coverage on a nearly daily basis. Hence MODIS might also provide a statistically better sample but the metrics shown in Fig. 7 and 8 are similar, except that ATSR underestimates by 0.07 and MODIS overestimates by a similar amount. It is noted that differences in the AOD value such as shown here between

ATSR and MODIS are also observed between the two MODIS instruments (with a 4 hour difference in overpass time which may influence the results) and between MODIS and MISR (Zhao et al., 2017; Zhang et al., 2017). MISR flies on Terra and has a swath width which is only a little smaller than that of ATSR. The comparison of the ATSR and MODIS/Terra AOD data sets over China with AERONET does not show a clear advantage for one of these and statistically the validation results are similar in spite of differences in AOD values. This conclusion

applies to areas where both ATSR and MODIS provide quality data, and hence over the bright surfaces in western China, such as the Taklimakan desert, the ATSR data cannot be used. Further development is needed to account for the surface effects on the ATSR TOA radiance. In conclusion, the MODIS/Terra and ATSR AOD are different, but there is no clear preference, as regards data quality, for one or the other. In view of the slight overestimation of MODIS and the slight underestimation of ATSR, the complimentary use of the AOD retrieved from these

instruments may provide added value for, e.g., data assimilation in chemical transport models.

The spatial distribution and the temporal behaviour of the ATSR and MODIS AOD data sets show similar features, with similar covariation in time and space. The spatial AOD distribution is a 20-year climatology which updates and extends earlier climatologies derived from MODIS C5.1 data, for different periods, usually starting in ca. 2000 (Terra, e.g., Guo et al., 2016; Luo et al., 2014) or 2002 (Aqua, e.g., He et al., 2016). The 3-D climatology

by Guo et al. (2016) focuses on the frequency of occurrence of aerosols over China for 2006-2014. MODIS C5.1 publications on AOD over China usually address certain aspects including, e.g., regional studies over SE China or over regions such as BTH (e.g., Li et al., 2007; Tang et al., 2016), YRD (e.g., Li et al., 2015; Kang et al., 2016) or PRD (e.g.,Bilal et al., 2017), differences between DT and DB retrieved AOD over certain regions, or validation. However, none of them presents an overview for all China, or even all Eastern China, and addresses differences

between different regions. CSHNET (Wang, Y., et al., 2011) or CARSNET (Che et al. 2015) do provide data all over China, however, these are point measurements with low coverage over rural areas. Satellite data fill these gaps but with lower temporal resolution.

As briefly discussed in Sect. 5, some distinct differences in the AOD seasonal behaviour can be observed over different parts of China, i.e. from south to north and from east to west. This forms the basis for a more detailed

study for selected areas over different parts of China (Sogacheva et al., in preparation).

The AOD time series presented in Sect. 6 are for all China, including both the relatively clean western China and the relatively polluted SE and SW China. Clearly, this is not a good representation to show differences of emission abatement policy for either aerosols (e.g., Zhao et al., 2017) or precursor gases (e.g., van der A et al., 2017) with a complicated effect on AOD (Lin et al., 2010), and their effects on different parts of industrialized China. In

addition, the evolution of the AOD is not only determined by policy and economic development but also by the evolution of living standards and migration of people in China, such as urbanization and development of agriculture which may be different across the country. Furthermore, different sources influence the aerosol content in different parts of China and in different seasons, i.e. dust emission in the west in the spring, biomass burning in the summer/autumn seasons in eastern China and the emissions of VOCs from vegetated areas (e.g. Tan et al.,

2015). The effects of these emissions are augmented by the meteorological conditions which also vary by region





and season (e.g., Ding and Murakami, 1994; Domros and Peng, 1988; Song et al., 2011; Jiang et al., 2015) and large scale periodic climate variability (e.g., Liu et al., 2013). Therefore, AOD time series will be further discussed by considering different regions in a different paper (Sogacheva et al., in preparation).

## 8. Conclusions

Two decades of satellite-derived aerosol optical properties provide an extended aerosol climatology over China (1995-2015), using the most recent retrieval results. The analysis of the data from different sensors shows that:

- The MODIS/Terra C6 DTDB merged AOD over China is distinctly higher than that retrieved form ATSR using the ADV v2.31 algorithm and the difference increases with increasing AOD.
- Validation of both data sets over China shows that both MODIS-Terra C6 DTDB and ATSR ADV v2.31
AOD compare well with AERONET reference data but MODIS slightly overestimates and ATSR slightly underestimates with respect to the AERONET AOD.
- AOD time series for ATSR and MODIS AOD show similar features and, although with a substantial offset, they provide complimentary information as regards the AOD increase in the late 1990s (pre-EOS) and the apparent decrease after the end of the ENVISAT mission in April 2012.
- Seasonal variations in the AOD are evident and vary for different parts of China.

The regional variation of seasonality and long-term behaviour of the AOD over China will be discussed in Sogacheva et al. (in preparation).

## Data availability

The ATSR data used in this manuscript are publicly available (after registration a password will be issued) at:
http://www.icare.univ-lille1.fr/. MODIS data are publicly available at: https://ladsweb.modaps.eosdis.nasa.gov/ and CALIOP data are avaialble via the LIVAS netcdf database (after obtaining login credentials) at ftp://lidar.space.noa.gr. A technical description of the LIVAS database is available under: http://lidar.space.noa.gr:8080/livas/. A brief description of the product can be found in the recently published article in ACP: (http://www.atmos-chem-phys.net/15/7127/2015/acp-15-7127-2015.html). AERONET data are
available at AERONET: https://aeronet.gsfc.nasa.gov/

## Acknowledgements

Work presented in this contribution was undertaken as part of the MarcoPolo project supported by the EU, FP7 SPACE Grant agreement no. 606953 and as part of the Globemission project ESA-ESRIN Data Users Element (DUE), project AO/1-6721/11/I-NB, and contributes to the ESA/MOST DRAGON4 program. The ATSR
algorithm (ADV/ASV) used in this work is improved with support from ESA as part of the Climate Change Initiative (CCI) project Aerosol_cci (ESA-ESRIN projects AO/1-6207/09/I-LG and ESRIN/400010987 4/14/1-NB). Further support was received from the Centre of Excellence in Atmospheric Science funded by the Finnish Academy of Sciences Excellence (project no. 272041). Many thanks are expressed to NASA Goddard Space Flight Center (GSFC) Level 1 and Atmosphere Archive and Distribution System (LAADS)
(http://ladsweb.nascom.nasa.gov) for making available the L3 MODIS/Terra C5.1 and C6 aerosol data.





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





**Figures and Tables**

**Figure 1.** The study area is mainland China, i.e. the area within the Chinese border, indicated in this elevation map by the blue line. Also indicated are some major cities (purple dots) and the locations of the AERONET sites (yellow diamonds) used in this study for validation. The colour of the dot inside each diamond gives an indication for the length of the data record at that site (red: years, green: months, blue: days). Areas mentioned in this paper are the Taklimakan Desert (TD), Gobi Desert (GD) and Tibetan Plateau (TP). Other areas are the Beijing/Tianjin/Hebei (BTH) area, the Yangtze River Delta (YRD) including Shanghai and Nanjing, the Pearl River Delta in the south including Guangzhou and the North China Plain (NCP) including BTH, YRD and the area in between.

**Figure 2.** Spatial distribution of the AOD over China, aggregated over the (full) years 2000-2011. The AOD has been retrieved from ATSR-2 and AATSR data, using the ADV algorithm v2.31. The AOD scale is presented in the colour bar to the right of the map. Areas for which no data are available are shown in white.

**Figure 3.** Spatial distribution of the MODIS/Terra C6 DTDB merged AOD over China, aggregated for the full years 2000-2011.

**Figure 4.** Difference between MODIS C6 and C5.1 merged DTDB AOD, see colour scale at the right. Red colours indicate that C6 is higher, blue colours indicate that C5.1 is higher.

**Figure 5.** Seasonal maps of AOD at 532 nm derived from CALIOP over the area 35º-45º N, 70º-150º E, including the Taklimakan/Gobi deserts and the Beijing-Tianjin-Hebei area, derived from 9 years of CALIPSO overpasses (2007-2015): winter (DJF), spring (MAM), summer (JJA) and autumn (SON).

**Figure 6.** Vertical distribution of the climatological extinction coefficient profiles at 532 nm over the area 35º-45º N, 70º-150º E, including the Taklimakan desert and the Beijing-Tianjin-Hebei area, derived from 9 years of CALIOP measurements (2007-2015): winter (DJF), spring (MAM), summer (JJA) and autumn (SON). The solid and broken lines indicate the mean and maximum elevation of the aerosol extinction above the surface.

**Figure 7.** Density scatterplot of ATSR-retrieved AOD, using ADV v2.31, over China for the years 2002-2012, versus AOD from AERONET stations in mainland China (cf. Fig. 1 and Table 1). The filled circles are the averaged ATSR AOD binned in 0.1 AERONET AOD intervals (0.2 for AERONET AOD>1.0) and the vertical lines on each circle represent the 1-sigma standard deviation of the fits. Statistics in the upper left corner indicate correlation coefficient r, bias, standard deviation, root mean square (rms) error and number of data points (N). The colour bar on the right indicates the number of data points.

**Figure 8:** Same as Fig. 7, but for MODIS/Terra C6, merged DTDB AOD data.

**Figure 9.** Difference of ATSR AOD minus MODIS/Terra C6 DTDB merged AOD, aggregated for the years 2000-2011, over China. The values in increments of 0.05 are given in the colour scale to the right.

**Figure 10.** MODIS/Terra C6 DTDB merged AOD versus AATSR ADV v2.31 AOD, for collocated AATSR-MODIS-AERONET data, as described in the text of Sect. 4.3. The colours, see scale at the right, indicate the difference between the MODIS/Terra and ATSR overpass times in minutes.

**Figure 11.** Seasonally averaged maps of the ATSR (left) and MODIS (right) retrieved AOD distribution over China for the years 2000-2011: winter (DJF), spring (MAM), summer (JJA) and autumn (SON). The AOD colour scale is on the right.

**Figure 12.** Time series of ATSR and MODIS retrieved AOD over China for two decades: 1995-2015. Note that data are missing in the beginning of the ATSR-2 observation period in 1995, and also MODIS/Terra data are available starting in April 2000; AATSR data start in May 2002. Therefore, the ATSR-2 1995 data point includes data for June-December and for 1996 July-December are included. ATSR-2 was used for 1995-7/2002, for 8/2002-2011 AATSR data were used. AATSR for 2012 is not shown since only 3 months of data are available, mainly in the winter (JFM) with low coverage. The MODIS 2000 data point includes data from April to December.