# Peer review of "Two decades of satellite observations of AOD over mainland China."

_Atmospheric Chemistry and Physics, 2017_

## Short Comment (SC1) · 19 Oct 2017

Hi,

It is a nice paper. I have a couple of questions about the CALIOP data you have used in the study.

1. Are the day-time and night-time data of CALIOP all used in this study ?

2. Did you consider the cloud aerosol discrimination (CAD) score of the aerosol extinction coefficients data? If so, what criteria did you apply to select the aerosol extinction coefficients?

3. What is the vertical range (lower troposphere? upper troposphere? lower strato-

sphere? or any other?) that you have considered when calculating the AOD, e.g., in Fig.5 ?

Thank you very much.

Regards, Xue

---

## Short Comment (SC2) · 20 Oct 2017

Dear Dr de Leeuw and co-authors,

**May the MODIS AOD seasonality be mislabelled in Fig. 11?** Comparison with the ADV AOD seasonality (left-hand column), CALIPSO AOD seasonality (Fig. 5 of manuscript), and Fig. 3 of Luo et al. (2014, doi:10.1002/joc.3728) suggests to me that this may be the case. In particular, the seasonality of high AOD over the Taklamakan Desert, BTH and other regions appears to be one season out of sync. For further details, please see the annotated figure included in this comment.

It is of course possible that these differences in seasonality between the different AOD products are genuine. But I suggest checking the code used to make Fig. 11, just in

case a mistake has been made.

Please do let me know if I have misunderstood anything.

Thank you,

With kind regards,

Benjamin

[Figure]

ADV · MODIS

Over the Taklimakan desert, CALIPSO AOD appears highest in MAM (see Fig. 5)

MAM?

Over ocean, ADV AOD appears highest in MAM

Are the MODIS seasons mislabelled?

In addition to comparing with the ADV AOD (left-hand side) and CALIPSO AOD results (Fig. 5), one can also compare with the MODIS AOD results of Luo et al. (2014; their Fig. 3).

JJA?

Over BTH and Indo-Gangetic plain, ADV appears highest in JJA

SON?

DJF?

Figure 11. Seasonally averaged maps of the ATSR (left) and MODIS (right) retrieved AOD distribution over China for the years 2000-2011: winter (DJF), spring (MAM), summer (JJA) and autumn (SON). The AOD colour scale is on the right.

**Fig. 1.** Copy of Fig. 11 with annotations added.

---

## Author Comment (AC1) · 20 Oct 2017

Dear Benjamin,

I am very grateful that you found this mistake: indeed I included the wrong figure. The correct Fig. 11 is included below. Since I drafted the text based on this correct Fig.11 it does not need changing.

I have contact the editorial staff to ask for advice on how to proceed, but unfortunately this cannot be changed in the discussion paper. Hence I hope that readers and reviewers will read your comment and our response and we will of course include the correct Fig. 11 in the final revised paper, after review has been completed.

Best regards,

Gerrit and co-authors.

Corrected Fig. 11:

---

## Author Comment (AC2) · 22 Oct 2017

Dear Xue Wu,

Thanks for your very quick comments, which help clarify what we did also to other readers. Please find the responses below.

1. Are the day-time and night-time data of CALIOP all used in this study?

Both daytime and nighttime CALIOP observations have been included in the analysis.

2. Did you consider the cloud aerosol discrimination (CAD) score of the aerosol extinction coefficients data? If so, what criteria did you apply to select the aerosol extinction coefficients?

The Winker et al. (2013) procedure and filtering criteria are followed. Specifically, regarding the CAD score, the methodology of CALIPSO L3 for ensuring the use of only cloud-free profiles is applied. According to this methodology only features having a CAD score between -100 and -20 are used. Profiles which do not fulfil the L3 CALIPSO CAD score criteria are filtered. We have followed the quality control procedures and filters described in the literature (Winker et al., 2013; Marinou et al., 2017).

3. What is the vertical range (lower troposphere? Upper troposphere? Lower stratosphere? Or any other?) that you have considered when calculating the AOD, e.g., in Fig.5 ?

The CALIPSO profiles of vertical extinction coefficient at 532 nm with 5 km horizontal resolution are used to calculate the AOD in 1x1 deg2 horizontal resolution (Winker et al., 2013; Amiridis et al., 2013). CALIPSO L2 profiles are initially horizontally averaged and then vertically integrated between the surface level and the lower stratosphere (~30km height).

Best regards, Gerrit, Manolis and co-authors

---

## Referee Comment (RC1) · A. M. Sayer (Referee) · 25 Oct 2017

General comments and recommendation

I am posting this review under my own name (Andrew Sayer) as I've discussed the work with the authors at a recent workshop, and also am a developer of the MODIS aerosol products used as part of the analysis. The study looks at 20 years of satellite AOD over China, using the ADV algorithm applied to ATSR2/AATSR measurements, as well as the MODIS Terra Deep Blue/Dark Target (DBDT) merged product. The two are non-negligibly offset but have similar spatial patterns and interannual variability. CALIOP data are also used (principally to look at the vertical profile of aerosol loading), and AERONET is used as a validation data source. This represents part one of an analysis,

with a second part about seasonality and trends to follow. (Incidentally, I would be very keen to review the follow-up by Sogacheva et al, which I refer to hereafter as 'part 2', when this is submitted.)

On the whole this is an interesting paper but I think as currently presented it suffers from being split into two parts, and would feel more complete if the seasonal analysis and trend analysis were part of this study. In general I am not a fan of annual analyses because, especially in this study region, the aerosol loading, type, and satellite sampling completeness (due to e.g. clouds, snow) both show strong seasonal variability. This is accentuated by the ATSRs having a swath width of around 550 km as compared to MODIS' 2330 km, as well as ATSR-2 temporary failures during parts of 1995/1996 meaning that seasons were missed out. As a result I don't think that comparing an annual mean AOD is very meaningful. It's impossible to tell without going deeper how much of the offset is driven by sampling differences. So I imagine this aspect will be addressed more in part 2. I understand that changing to a seasonal focus here would make the combined paper too long and render the follow-up part 2 obsolete. In that case, this first paper could include more of a perspective on evaluating and thinking about how to merge data records. The ATSRs and Terra (MODIS and MISR) are among the few long-term morning aerosol records; the paper does a nice job of showing the offset (on a country-level scale and in spatial patterns) between them, but similarity in year-to-year variations, so perhaps the authors could go deeper here and identify for example when/where it is sensible to consider the records 'consistent' (and pave the way for a future attempt to merge records) based on a more comprehensive evaluation of the data sets vs. AERONET.

My recommendations are therefore for revision and re-review. I think this is an interesting paper but could have a better focus and clearer split between this and the follow-on part 2 paper. Indeed, after my initial read and talking to the authors, many of my suggestions were things they had planned for part 2. So I think this study would benefit from a clearer definition of scope, with this part 1 setting up the data, evaluating, and

giving a big picture, and part 2 focusing more directly on seasonality and time series. I have some specific comments and suggestions to this end below. While this is a long review, that should not be taken as a negative statement about the paper, but rather my feeling that this is an interesting and important topic which deserves careful discussion.

Specific comments

Title: from my reading of the manuscript and discussion with the authors, this paper does seem to be quite tightly linked with the forthcoming Sogacheva et al study. I therefore wonder if this could be reflected in the title? For example, add something like "Part 1: large-scale patterns and data set evaluation". That way the reader will immediately be alerted that there is a second related study they should look at too.

Abstract: this should be more concise (ideally one paragraph). I suggest a shortening; for example, the first two sentences could be deleted. The second paragraph could also be more concise, for example saying that ADV covers 1995-2012 and MODIS Terra 2000 onwards (and up to 2015 used in this paper). I'd delete the comment at the end about decreasing AOD since 2012 since trends/variations are not quantitatively discussed in the manuscript.

Page 3, lines 6-19: the authors state that the "focus" of the data is from the combined ATSR-2/AATSR ADV record, and it is "further extended to 2015" using MODIS Terra. I appreciate that this may in part be because the authors are developers of ADV and perhaps the analysis started that way. However as a reader it appears the analysis evolved and roughly equal emphasis and weight is given to each of ADV and MODIS, with CALIPSO being the add-on.

Page 4, first paragraph: some of the text here is repetitive with that on the previous page (talking about a 1995-2012 ADV record extended with MODIS) and can be short-ened/deleted.

Page 4, line 13: this study uses the merged Deep Blue/Dark Target MODIS product.

The Levy et al (2013) paper cited here focuses on the Dark Target land and ocean algorithm changes, with a brief section about the DBDT merge. Sayer et al (2014), which is cited elsewhere, has an expanded description (and evaluation) of the DBDT merge (which was new to Collection 6). Hsu et al (2013), also cited elsewhere, is about Deep Blue updates. To better direct the interested reader to information about what changed in Collection 6 I'd suggest citing all 3 papers here, or else citing only the Sayer 2014 paper, rather than Levy 2013 alone.

Page 4, line 25: This Levy 2013 reference would also in my view be better off as Sayer 2014 as well/instead. (I'd argue that I am not biased in this since Rob Levy and I are authors on both papers. Levy 2013 is really mostly about Dark Target while Sayer 2014 is more explicitly about Deep Blue, Dark Target, and the merge all together.)

Section 2: Another obvious choice for inclusion in this study would be the MISR aerosol product, which is also on Terra (i.e. same local time and time series length) but has a different measurement and retrieval type from MODIS, and a swath more similar to ATSR. Or SeaWiFS, which covered late 1997 to 2010 but had an early afternoon crossing time. At present later on the authors compare ATSR and MODIS and note some similarities and some differences in spatial patterns and AOD magnitude. The reasons for using ATSR and MODIS are a bit arbitrary so adding an additional data set (I would probably suggest MISR) would help strengthen the understanding of where the satellite products agree and where they do not (for either algorithmic or sampling reasons). I realise this is not a negligible amount of work but do think that adding a third perspective will be useful.

Page 6 line 1: the new calibration update described in Levy et al (2013, 2015) and Lyapustin et al (2014) was not applied in Collection 6 Dark Target data, so this sentence is incorrect and should be deleted. Deep Blue does include polarisation, gain, and response-vs-scan corrections and detrending for Terra (which are described in the Sayer et al 2015 reference cited in the next sentence). But Dark Target does not, and the Levy/Lyapustin references are not related to what is done in Deep Blue. I suggest

rewording to reflect this.

Section 2.1.3: it is not clear to me from reading this which specific CALIPSO product and data version are used here. It says the 'L2 product', but there are several of those (aerosol profile, aerosol layer, combined cloud/aerosol). Also, is the latest version 4? If not, it should ideally be updated, due to some improvements in the version 4 product to do with calibration, layer detection/extent, lidar ratios, and type classification.

Page 7 line 5: the AERONET team like use of the level 1.5/2.0 products to have a reference to their cloud screening and QA paper, Smirnov et al (2000, doi:10.1016/S0034-4257(00)00109-7).

Page 9 line 5: MODIS Collection 6.1 data have recently been released; at the time of writing, Terra data from 2000-2008 are freely available and the reprocessing up to the end of the authors' 2015 study period should be complete by mid-December. The C6/C6.1 changes in the Deep Blue and Dark Target land algorithms are not negligible. For example Deep Blue made some improvements over mountains and biomass burning aerosols, and Dark Target made some improvements over cities. All of these feature in the study region. Note that the merging logic in C6.1 is unchanged. C6.1 is expected to remain the current standard version for the coming several years. In an ideal world this analysis would therefore be performed with C6.1 data; I do realise though that repeating the analysis takes time. To start with I have attached a figure (as a Supplement) showing the multiannual monthly mean C6.1 Terra merged AOD and C6.1-C6 AOD difference for the period available today (2000-2008), for the authors' study region. As you can see there are some pretty significant systematic differences, e.g. over the Tibetan Plateau (TP) and mountains to the east of that in particular, but elsewhere as well (systematic decrease over the North China Plain in some months). Since the TP is one of the areas highlighted for regional analysis and where ATSR and MODIS (C6) climatologies were different, this significant change over the TP in C6.1 (decrease by more than 0.25 in some areas) is relevant to the analysis and discussion. Note in the attached figure the colour scales are truncated, i.e. some AODs are

above 1 and some differences larger than 0.25. As a result I do think that changing to the latest C6.1 would be beneficial. Otherwise the authors' analysis will shortly seem outdated; using C6.1 would instead make it really at the forefront. We don't yet have publications about the C6.1 changes, but user guides illustrating some changes are online at https://modis-atmosphere.gsfc.nasa.gov/documentation/collection-61 .

Section 3: As noted earlier, I do not think that annual/multiannual averages are particularly meaningful because of strong seasonal variation in AOD, aerosol loading, and sampling. However I do understand that the authors want to keep most of the seasonal analysis for the follow-up paper, and they do give caution about sampling etc in the discussion here. But to give a hint of where this may be important, Figures 2 and 3 could perhaps be expanded to include a little more information. For example, panels showing the number/fraction of months corresponding to the aggregates, the variability within a year (i.e. the mean over all years of the standard deviation of AOD from Jan-Dec), the interannual variability (i.e. the standard deviation of annual aggregates), etc. I think this would provide better context for the discussion of spatial features between ATSR and MODIS data here. As mentioned in my comment about section 2, suggesting inclusion of MISR data, if you're trying to assess whether the aggregates are giving the same picture or not then the more (relatively) independent data sets you can look at, and the more you can see where sampling is good and where it isn't, the better.

Page 11 lines 8-15: the authors created a merged MODIS AOD product based on Collection 5.1 data, to compare with that provided in Collection 6 (as no official merge was provided in Collection 5.1). This is a nice thing to see as we as an algorithm team have not directly published C5.1/C6 difference maps from the merged product. However at present this part is not well integrated into the paper. The text says that "These differences will be further discussed after evaluation of the satellite AOD products versus available AERONET reference AOD data.", but I could not find the further discussion in the paper. Was this removed in a draft, perhaps? If the authors want to show C5.1/C6 differences then my suggestion would be to add more discussion about how these

changes may affect previously-published studies and their conclusions. Otherwise it can probably be deleted since C5.1 is considered obsolete now.

Section 3.1.3: this section feels a bit disjointed from the rest. For example, why do we only have the thin strip of CALIPSO data from 35-45 N in Figure 5 rather than the whole study area? I think that better use could be made of CALIPSO's strengths here as well. For example it's good to see the seasonal vertical profile of extinction in Figure 6 across this strip, but showing something like depolarisation ratio or colour ratio or even aerosol type frequency would allow us to see the vertical structure and changing seasonal nature of aerosol composition, which we can't see from the extinction profile alone. I might also consider moving this part to fit in with the seasonal discussion near Figure 11.

Section 4: I like the validation shown, and the comparison between MODIS and ADV when both are collocated with AERONET in Figure 10. It would be good to add some statistics to Figure 10, similar to the other panels. Since the AERONET sites cover a range of aerosol and surface conditions, I think it would also be instructive to further separate the points in these figures to different regimes (e.g. compare dust/pollution dominated scenes, classified using AERONET AOD and Ångström exponent, maybe split sites by surface type). There is clearly variation in surface cover and aerosol type, and we know that algorithm errors are dependent on those factors, so it makes sense to look at the data that way. For example, in Figures 7 and 8 there are some high-AOD points with high biases and low biases. It may be that high are associated with one aerosol type and low another, or it may be that the data are unbiased but noisy. With the analysis as presented, we don't know. Understanding if there are systematic surface/aerosol type dependent errors in the ATSR and MODIS data here would also help understand the differences in seasonal and multiannual aggregates presented in the paper and then aid in the interpretation of the follow on part 2 paper. (I realise, and the authors state, that most of the AERONET sites are in low-lying vegetated/suburban/urban areas, rather than deserts, so splitting by surface type may not

be feasible. But doing a simple cut based on AERONET AOD and Ångström expo-
nent to separate into background, fine-mode dominated, and coarse-mode dominated
aerosols should be possible.)

Section 5: although most of the seasonal analysis is to come in the follow up part 2
paper, some of it is included here. Figure 11 is very interesting (I saw also the on-
line comment containing the correct seasonal panel ordering) as it reveals the large
seasonality, going back to my earlier comment about multiannual aggregates alone
not being useful. I think this figure though is also illustrative of my earlier point about
needing to understand differences rather than just state them. Clearly both sensors
see similar patterns and seasonality but there are offsets (and again the differences in
C6.1 are relevant here). Looking at type-dependence of errors vs. AERONET (sug-
gested above) helps answer this question. Adding different measurement types also
helps. Right now we have ATSR (dual-view narrow swath) and MODIS (single-view
broad swath). I think the authors should move the CALIPSO seasonal maps (currently
Figure 5) to be an extra column of this Figure, and expand them to cover the whole
spatial domain. That provides another different measurement technique (active curtain
profiling by lidar). Adding MISR as a fourth would also help: it is multi-view like AATSR,
but has more angles (9 vs. 2), and a different spectral range, which provides different
information content. Put them all together and you have a nice ensemble of 4 different
seasonal climatologies from different sensors and retrieval techniques, and you will re-
ally begin to see where the data are consistent and where they are not. That tells data
users where to be cautious and data producers where to focus effort.

Figure 11: there are some strange artefacts near the northern end of the MODIS DJF
panel (appears as SON in the Discussion paper). Are these really areas of zero AOD,
or are these areas without valid retrievals, which should then appear white for missing
data? I think it is more likely this is missing data; this should be checked. There's a
similar smaller artefact along the Himalayas in the MODIS MAM/JJA panels (appear
as DJF/MAM in the Discussion papers), near 35 N 80 E.

Section 6: I have mixed feelings about this section. I understand that temporal variations are one of the main topics of the follow on part 2 study by Sogacheva et al. Here the authors show that MODIS and ATSR time series are offset but have similar inter-annual variations. However these are annual aggregates across the whole of China so again the question is raised of how meaningful these numbers are (they are not representative of any one place in China or season so it is not clear what I would want to use these numbers to assess or calculate). In my view including it here opens a bit of a can of worms, while it is interesting to see. My suggestion is to remove Section 6 and save all discussion of time series and trends for the follow on paper. It is already fairly well signposted that seasonality and trends are the topics for the follow on, so I don't know that it is necessary to introduce that topic here. The end of the conclusion can be cut down to do this signposting as well.

Section 8: I think Sections 7 and 8 can be condensed into a single discussion and conclusion section. I don't see what is gained by having the separate section 8 since it is short, and most of what is in section 7 is fitting for a conclusion. The two are even somewhat repetitive, e.g. last sections mentioning time series will be discussed in Sogacheva et al. Condensing this down, as well as saving Section 6 for the follow-on, will help prevent the paper from getting too long if the analysis is expanded to cover some of my suggestions.

Please also note the supplement to this comment:
https://www.atmos-chem-phys-discuss.net/acp-2017-838/acp-2017-838-RC1-supplement.pdf

---

## Referee Comment (RC2) · Anonymous Referee #1 · 15 Nov 2017

De Leeuw et al. reported a new 3-D climatology of AOD over mainland China based on the combination of MODIS, ATSR and AERONET. Compared with MODIS, the radiometer of ATSR extends the AOD time series albeit the large offset between these two products. This topic is not new, but the methods used are robust and analysis is sound. For instance, ground-based AERONET AODs have been used to validate the space-borne AOD, in addition to the intercomparisons between MODIS AOD and ATSR AOD. Overall, the paper is well written, and deserves publications in ACP if the following concerns being successfully addressed.

**General comments:**

1. Table 1: It is well known that the time period when AERONET stations operated in the past varies a lot by regions. The time period for each site in Table is suggested to be added so that the readers can be better informed. Likewise, the various time periods or AREONET sites can be interpreted by differentiating the scatters in Figs. 7 and 8 in various shapes. This will help give more information with regard to the validation results.

2. Why only CALIOP AOD over north parts of mainland China is shown in Fig. 5? The topic is the AOD climatology over China, and it will be useful to show CALIOP AOD throughout the study area. Meanwhile, you can highlight in Fig. 5 the region of interest (35-45 N; 70-150E) for the cross-section map in Fig.6.

**Minor comments:**

1. Page 2, line 22-23: The following references could be considered to be added to inform the readers of the background of aerosol effect. "Aerosol particles are important because of their effects on weather and climate (e.g., Rosenfeld et al., 2008; Koren et al., 2014; Guo et al., 2016), health (Pope et al., 2009; Anenberg et al., 2010), atmospheric chemistry, visibility(Sisler and Malm, 1994), cultural heritage, etc."

References:
Anenberg, SC, Horowitz, LW and Tong, DQ *et al*. An estimate of the global burden of anthropogenic ozone and fine particulate matter on premature human mortality using atmospheric modeling. Environmental Health Perspectives 2010; 118:1189-1195.
Guo, J., M. Deng, S. S. Lee, F. Wang, Z. Li, P. Zhai, H. Liu, W. Lv, W. Yao, and X. Li (2016), Delaying precipitation and lightning by air pollution over the Pearl River Delta. Part I:

Observational analyses, J. Geophys. Res. Atmos., 121, 6472–6488, doi:10.1002/2015JD023257.

Koren, I., G. Dagan, and O. Altaratz, 2014. From aerosol-limited to invigoration of warm convective clouds, Science, 344(6188), 1143–1146, doi:10.1126/science.1252595.

Pope C. A., M. Ezzati, and D.W. Dockery. 2009. Fine-particulate air pollution and life expectancy in the United States. N. Engl. J. Med. 360: 376-386.

Rosenfeld, D., U. Lohmann, G. B. Raga, C. D. O'Dowd, M. Kulmala, S. Fuzzi, A. Reissell, and M. O. Andreae (2008), Flood or drought: How do aerosols affect precipitation?, Science, 321(5894), 1309–1313.

Sisler, J. F., and Malm, W. C. (1994). The relative importance of soluble aerosols to spatial and seasonal trends of impaired visibility in the United States. Atmospheric environment, 28(5), 851-862.

2. Page 17, line 14: "Varying between -20 and 80 minutes"? Also, the discussion as regards the overestimation of MODIS AOD relative to ATSR AOD could be problematic. In the morning, the boundary layer height (BLH) generally increases as a result of increasing turbulence caused by increasing incoming solar radiation (Guo et al., 2016; Stull et al., 1988; Petäjä et al., 2016). The later overpass time of MODIS/Terra will have higher BLHs, resulting in lower aerosol concentrations and smaller AOD. This is opposite to the results shown in Fig.10. Therefore, more insightful discussion is much needed here (Li et al., 2017).

References:

Guo, J., Miao, Y., Zhang, Y., Liu, H., Li, Z., Zhang, W., He, J., Lou, M., Yan, Y., Bian, L., and Zhai, P.: The climatology of planetary boundary layer height in China derived from radiosonde and reanalysis data, Atmos. Chem. Phys., 16, 13309-13319, doi:10.5194/acp-16-13309-2016, 2016.

Li Z., J. Guo, A. Ding, H. Liao, J. Liu, Y. Sun, T. Wang, H. Xue, H. Zhang, B. Zhu, 2017. Aerosol and boundary-layer interactions and impact on air quality, National Science Review, doi: 10.1093/nsr/nwx117.

Petäjä, T, Jarvi, L and Kerminen, VM et al. Enhanced air pollution via aerosol-boundary layer feedback in China. Scientific Reports 2016; 6: 18998.

Stull, RB. An Introduction to Boundary Layer Meteorology, edited by R. B. Stull, Springer Netherlands, Dordrecht 1988.

---

## Short Comment (SC3) · 16 Nov 2017

Dear Gerrit and co-authors,

I am glad that the previous comment was helpful. Thank you for providing the corrected figure.

On another note, the following paper (published yesterday) may be a relevant reference:

Wang et al. (2017), Trends and Variability in Aerosol Optical Depth over North China from MODIS C6 Aerosol Products during 2001–2016, Atmosphere, https://doi.org/10.3390/atmos8110223

[Figure]

With kind regards,

Benjamin
* * *

---

## Short Comment (SC4) · 16 Nov 2017

Dear Benjamin,

Thanks a lot for this very useful reference! This paper by Wang et al. indeed relates strongly to part2 of our manuscript, which however, covers not only North China but also other areas over mainland China. We expect to have this part 2 ready for initial submission pretty soon!

Best regards, Gerrit
* * *

---

## Author Comment (AC3) · 18 Dec 2017

A. M. Sayer (Referee)

andrew.sayer@nasa.gov

**General comments and recommendation**

**I am posting this review under my own name (Andrew Sayer) as I've discussed the work with the authors at a recent workshop, and also am a developer of the MODIS aerosol products used as part of the analysis. The study looks at 20 years of satellite AOD over China, using the ADV algorithm applied to ATSR2/AATSR measurements, as well as the MODIS Terra Deep Blue/Dark Target (DBDT) merged product. The two are non- negligibly offset but have similar spatial patterns and interannual variability. CALIOP data are also used (principally to look at the vertical profile of aerosol loading), and AERONET is used as a validation data source. This represents part one of an analysis, with a second part about seasonality and trends to follow. (Incidentally, I would be very keen to review the follow-up by Sogacheva et al, which I refer to hereafter as 'part 2', when this is submitted.)**

We will recommend you as a reviewer when we submit part 2

**On the whole this is an interesting paper but I think as currently presented it suffers from being split into two parts, and would feel more complete if the seasonal analysis and trend analysis were part of this study. In general I am not a fan of annual analyses because, especially in this study region, the aerosol loading, type, and satellite sampling completeness (due to e.g. clouds, snow) both show strong seasonal variability. This is accentuated by the ATSRs having a swath width of around 550 km as compared to MODIS' 2330 km, as well as ATSR-2 temporary failures during parts of 1995/1996 meaning that seasons were missed out. As a result I don't think that com- paring an annual mean AOD is very meaningful. It's impossible to tell without going deeper how much of the offset is driven by sampling differences. So I imagine this aspect will be addressed more in part 2. I understand that changing to a seasonal focus here would make the combined paper too long and render the follow-up part 2 obsolete. In that case, this first paper could include more of a perspective on evaluating and thinking about how to merge data records. The ATSRs and Terra (MODIS and MISR) are among the few long-term morning aerosol records; the paper does a nice job of showing the offset (on a country-level scale and in spatial patterns) between them, but similarity in year-to-year variations, so perhaps the authors could go deeper here and identify for example when/where it is sensible to consider the records 'consistent' (and pave the way for a future attempt to merge records) based on a more comprehensive evaluation of the data sets vs. AERONET.**

We much appreciate these thoughts which in fact reflect to a large extent our way of thinking while developing this MS (or the set of 2 papers, if you like). In fact this MS started out as one. However, in the course of writing up the results, the data showed a multitude of aspects, such as the strong seasonality with AOD maxima in Spring and Summer while minima were observed in the winter, opposite to ground-based aerosol observations which peak in winter, the change in seasonal behavior from south to north, together with increasing AOD over time and possible decrease in recent years. And all of these show regional differences. This, together with the notice that ATSR and MODIS show similar behavior, both in space and time, but that AOD values are offset, with MODIS higher than ATSR, we also felt hat a data comparison was needed with AERONET as reference data set. Since neither ATSR nor MODIS appeared to be better

everywhere, actually in a statistical sense the difference with AERONET is similar but of opposite sign, there is no reason to disregard one of the two, which justifies combining them. Thus indeed it is tempting to think about merging, but we do not do that here. Indeed also sampling differences may be a problem, however, the similar behavior in the comparisons with AERONET, with a 5 times smaller data set from ATSR than from MODIS (as expected based on swath width) gave us confidence that both sensors can be used together. This was further confirmed by the similar behavior of the time series in the overlapping period. At the same time we strictly avoided looking at trends and only look at variations over time and try to identify why such changes might occur. With all of this, we soon came to the conclusion that all material together would make this MS very long and decided to split it into two papers with in paper 1 a focus on the description of the data sets, differences and validation, all with AERONET as reference, and more detailed analysis in paper 2 where indeed we will present differences between regions, the seasonality for each of these regions as well as long term variations of AOD over each of them.

Initially we also planned to submit both papers together. However, more work appears to be needed for paper2 and we see no reason for holding up the first paper while continuing to work on paper2, since both papers should be stand alone to begin with.

The second point is the annual analysis. We do not quite understand the objections since this seems to be common practice also in other recent publications of the temporal variation of AOD over China, e.g., Zhao et al. (2017).

**My recommendations are therefore for revision and re-review. I think this is an interesting paper but could have a better focus and clearer split between this and the follow-on part 2 paper. Indeed, after my initial read and talking to the authors, many of my suggestions were things they had planned for part 2. So I think this study would benefit from a clearer definition of scope, with this part 1 setting up the data, evaluating, and giving a big picture, and part 2 focusing more directly on seasonality and time series. I have some specific comments and suggestions to this end below. While this is a long review, that should not be taken as a negative statement about the paper, but rather my feeling that this is an interesting and important topic which deserves careful discussion.**

We thank the reviewer for this positive statement and his constructive comments. This split is indeed what we planned to do from the beginning and actually is done. However it may not have been clearly stated so at the end of the introduction we added a statement on the distribution of the topics discussed in paper 2 ("In a second paper (Sogacheva et al., in preparation) the seasonal and long-term variations for different regions in China are presented."), which at the same time allows for the removal of some of the references to paper2 later in the text.

**Specific comments**

**Title: from my reading of the manuscript and discussion with the authors, this paper does seem to be quite tightly linked with the forthcoming Sogacheva et al study. I therefore wonder if this could be reflected in the title? For example, add something like "Part 1: large-scale patterns and data set evaluation". That way the reader will immediately be alerted that there is a second related study they should look at too.**

Adding part1 and part2 would suggest that these papers are companion papers appearing in the same issue. However, paper2 is expected to be ready for submission in a few months and hence, assuming that with the current revision this MS will now be accepted for publication, the publication of both papers will be about half year apart. Therefore, we consider these papers as stand-alone papers, which have their own merit. However, we do agree with the reviewer to make the title more specific and add "using ATSR-2, AATSR and MODIS/TERRA: data set evaluation and large scale patterns".

**Abstract: this should be more concise (ideally one paragraph). I suggest a shortening; for example, the first two sentences could be deleted. The second paragraph could also be more concise, for example saying that ADV covers 1995-2012 and MODIS Terra 2000 onwards (and up to 2015 used in this paper). I'd delete the comment at the end about decreasing AOD since 2012 since trends/variations are not quantitatively discussed in the manuscript.**

Here we followed instructions on the ACP website "The abstract should be intelligible to the general reader without reference to the text. After a brief introduction of the topic, the summary recapitulates the key points of the article and mentions possible directions for prospective research". So we prefer to keep the first two sentences as general introduction. The second para summarizes our way of thinking and why we use MODIS

in addition to ATSR which is the principal data set used in this paper. The final comment is a summary of Sect. 6 where we discuss the qualitative decrease of the AOD. We keep the introduction as it is.

**Page 3, lines 6-19: the authors state that the "focus" of the data is from the combined ATSR-2/AATSR ADV record, and it is "further extended to 2015" using MODIS Terra. I appreciate that this may in part be because the authors are developers of ADV and perhaps the analysis started that way. However as a reader it appears the analysis evolved and roughly equal emphasis and weight is given to each of ADV and MODIS, with CALIPSO being the add-on.**

This is indeed true, as also stated below Fig. 1, and some of us are developers of ADV. However, we don't use the equal weight given to both data sets in this paper, since we don't merge them. We only present both of them and compare them. Furthermore, here we present our way of thinking: we started out with ATSR, not only as developers of ADV, but also as users of the data set. However, since AATSR was lost in April 2012, we were curious how the AOD evolved there-after and decided to use MODIS data for that purpose. Because of the long overlap period we could evaluate whether this is justified. This is what we write and we see no reason to change the text.

**Page 4, first paragraph: some of the text here is repetitive with that on the previous page (talking about a 1995-2012 ADV record extended with MODIS) and can be shortened/deleted.**

This text follows on from that on the previous page, where we briefly write what we do, i.e. which data sets are used, and here we describe the reason why the ATSR data set was extended to later years. We don't see a reason to change this text.

**Page 4, line 13: this study uses the merged Deep Blue/Dark Target MODIS product. The Levy et al (2013) paper cited here focuses on the Dark Target land and ocean algorithm changes, with a brief section about the DBDT merge. Sayer et al (2014), which is cited elsewhere, has an expanded description (and evaluation) of the DBDT merge (which was new to Collection 6). Hsu et al (2013), also cited elsewhere, is about Deep Blue updates. To better direct the interested reader to information about what changed in Collection 6 I'd suggest citing all 3 papers here, or else citing only the Sayer 2014 paper, rather than Levy 2013 alone.**

These references have been added with text as follows: "with further specification for DB updates by Hsu et al. (2013) and expanded description and evaluation of DTDB by Sayer et al. (2014)"

**Page 4, line 25: This Levy 2013 reference would also in my view be better off as Sayer 2014 as well/instead. (I'd argue that I am not biased in this since Rob Levy and I are authors on both papers. Levy 2013 is really mostly about Dark Target while Sayer 2014 is more explicitly about Deep Blue, Dark Target, and the merge all together.)**

Thanks for the suggestion, the reference has been replaced.

**Section 2: Another obvious choice for inclusion in this study would be the MISR aerosol product, which is also on Terra (i.e. same local time and time series length) but has a different measurement and retrieval type from MODIS, and a swath more similar to ATSR. Or SeaWiFS, which covered late 1997 to 2010 but had an early afternoon crossing time. At present later on the authors compare ATSR and MODIS and note some similarities and some differences in spatial patterns and AOD magnitude. The reasons for using ATSR and MODIS are a bit arbitrary so adding an additional data set (I would probably suggest MISR) would help strengthen the understanding of where the satellite products agree and where they do not (for either algorithmic or sampling reasons). I realise this is not a negligible amount of work but do think that adding a third perspective will be useful.**

The use of MISR could indeed have been an alternative choice, but SeaWiFS certainly not since the reason for adding another data set was the extension after April 2012, which is not covered by SeaWiFS. We did choose MODIS however, because this study was set out as part of the MarcoPolo project (see acknowledgements) in which our Greek colleagues (see list of co-authors) were working with MODIS and the Finnish colleagues with ATSR. Furhermore, MODIS is an obvious choice because it is by far the most frequently used data set.

It is further noted that MISR and MODIS data over China were discussed by Zhao et al (2017) who show in their Fig.2 (right panel) that MISR and MODIS show qualitatively similar trends, but MISR AOD over ECC is about 0.2-0.3 lower than that of MODIS/TERRA. We have not made a comparison between MISR and ATSR over China, but it seems that MISR would be lower than ATSR and thus the deviation from

AERONET would be larger. Furthermore, although indeed the swath widths of MISR and ATSR are similar, this does not mean that the swaths overlap. The slightly smaller MISR swath (than ATSR), and the likely collocation mismatch between the two instruments, may cause additional problems. Nevertheless, it could be interesting to follow up on the reviewer's suggestion and compare different instruments such as MODIS, MISR, SeaWiFS and ATSR, but this is out of scope for the current paper.

**Page 6 line 1: the new calibration update described in Levy et al (2013, 2015) and Lyapustin et al (2014) was not applied in Collection 6 Dark Target data, so this sentence is incorrect and should be deleted. Deep Blue does include polarisation, gain, and response-vs-scan corrections and detrending for Terra (which are described in the Sayer et al 2015 reference cited in the next sentence). But Dark Target does not, and the Levy/Lyapustin references are not related to what is done in Deep Blue. I suggest rewording to reflect this.**

This sentence has been removed and another sentence has been added to reflect the different corrections for DB and DT as suggested by the reviewer. The relevant text now reads: "Several changes have been made in C6 compared to C5.1. Details about these updates in C6 DT and DB data can be found in a number of recent studies (e.g. Levy et al., 2013; Tao et al., 2015; Sayer et al., 2015; Georgoulias et al., 2016). Corrections for polarisation, gain, and response-vs-scan corrections and detrending for MODIS/Terra degradation have been included in DB, but not in in DT. Another important update in C6 is the inclusion of a merged (DT and DB) dataset as described in Levy et al. (2013).": The reference to Lyapustin has bene removed from the reference list.

**Section 2.1.3: it is not clear to me from reading this which specific CALIPSO product and data version are used here. It says the 'L2 product', but there are several of those (aerosol profile, aerosol layer, combined cloud/aerosol). Also, is the latest version 4? If not, it should ideally be updated, due to some improvements in the version 4 product to do with calibration, layer detection/extent, lidar ratios, and type classification.**

We thank the reviewer for this constructive comment. At the end of Sect. 2.1.3 we have added "In this paper the ESA-LIVAS ("LIdar climatology of Vertical Aerosol Structure for space-based lidar simulation studies" project) database is used. LIVAS is developed based on CALIPSO V3 L2 Aerosol and Cloud Profile products towards a global multiwavelength (355, 532, 1064, 1570 and 2050 nm) aerosol and cloud optical database on a uniform 1°x1° grid resolution (Amiridis et al., 2015). Here the CALIPSO-based ESA-LIVAS product is used to provide the three-dimensional climatology of the aerosol distribution over China, for the period 01/2007-12/2015."

Regarding the CALIPSO version, indeed many changes have been applied from V3 to V4, including the change in the nighttime calibration algorithm (change of calibration height from 30-34 km to 36-39 km (Jayanta Kar et al, 2017), improved LRs and classification, improvements in the detection of tenuous aerosol layers (Travis D. Toth et al., 2017). The authors are aware of the improvements on CALIPSO V4 and already work towards the upgrade of the ESA-LIVAS database, though due to the computation resources and time needed, the upgraded database will be available for future studies.

**Page 7 line 5: the AERONET team like use of the level 1.5/2.0 products to have a reference to their cloud screening and QA paper, Smirnov et al (2000, doi:10.1016/S0034- 4257(00)00109-7).**

Thanks for this advice: reference added

**Page 9 line 5: MODIS Collection 6.1 data have recently been released; at the time of writing, Terra data from 2000-2008 are freely available and the reprocessing up to the end of the authors' 2015 study period should be complete by mid-December. The C6/C6.1 changes in the Deep Blue and Dark Target land algorithms are not negligible. For example Deep Blue made some improvements over mountains and biomass burning aerosols, and Dark Target made some improvements over cities. All of these feature in the study region. Note that the merging logic in C6.1 is unchanged. C6.1 is expected to remain the current standard version for the coming several years. In an ideal world this analysis would therefore be performed with C6.1 data; I do realise though that repeating the analysis takes time. To start with I have attached a figure (as a Supplement) showing the multiannual monthly mean C6.1 Terra merged AOD and C6.1-C6 AOD difference for the period available today (2000-2008), for the authors' study region. As you can see there are some pretty significant systematic differences, e.g. over the Tibetan Plateau (TP) and mountains to the east of that in particular, but elsewhere as well (systematic decrease over the North China Plain in some months). Since the TP is one of the areas**

**highlighted for regional analysis and where ATSR and MODIS (C6) climatologies were different, this significant change over the TP in C6.1 (decrease by more than 0.25 in some areas) is relevant to the analysis and discussion. Note in the attached figure the colour scales are truncated, i.e. some AODs are above 1 and some differences larger than 0.25. As a result I do think that changing to the latest C6.1 would be beneficial. Otherwise the authors' analysis will shortly seem outdated; using C6.1 would instead make it really at the forefront. We don't yet have publications about the C6.1 changes, but user guides illustrating some changes are online at https://modis-atmosphere.gsfc.nasa.gov/documentation/collection-61 .**

We thank the reviewer for sharing this information and the useful advice. However, we started this study several years ago, as part of the MarcoPolo project which ended in March 2017. We learned about a MODIS update to C6.1 only during the AEROCOM meeting in October 2017, i.e. after the submission of this paper. C6.1 reprocessing had not been finished. Hence it is not reasonable to ask us at this stage to redo our analyses. Furthermore, the material provided by the reviewer shows that the change from C6 to C6.1 has a large effect over the TP, but it is minor over most of the rest of China. In addition, without further knowledge and additional validation, documented in peer-reviewed publications, it would be hard to justify and explain the use of C6.1. However, to alert the reader of the upcoming C6.1 and differences with C6 which affects the results of the current study, we have added the following sentence above Fig. 9: "It should be noted here that, as pointed out by a reviewer (A. Sayer, private communication, Nov 2017), the MODIS team works on a new version, Collection 6.1, which has significantly lower AOD over, e.g. the Tibetan Plateau, while over east China the differences between C6 and C6.1 are very small."

**Section 3: As noted earlier, I do not think that annual/multiannual averages are particularly meaningful because of strong seasonal variation in AOD, aerosol loading, and sampling. However I do understand that the authors want to keep most of the seasonal analysis for the follow-up paper, and they do give caution about sampling etc in the discussion here. But to give a hint of where this may be important, Figures 2 and 3 could perhaps be expanded to include a little more information. For example, panels showing the number/fraction of months corresponding to the aggregates, the variability within a year (i.e. the mean over all years of the standard deviation of AOD from Jan-Dec), the interannual variability (i.e. the standard deviation of annual aggregates), etc. I think this would provide better context for the discussion of spatial features between ATSR and MODIS data here. As mentioned in my comment about section 2, suggesting inclusion of MISR data, if you're trying to assess whether the aggregates are giving the same picture or not then the more (relatively) independent data sets you can look at, and the more you can see where sampling is good and where it isn't, the better.**

We have responded to the use of annual averages and the use of MISR in the above.

As regards the statistics, we have first computed monthly averages and then used these 12 data points to compute the yearly averages. Obviously, over bright surfaces, especially in the winter months, there are data missing over certain areas in the north and west of China, and there the number of months contributing to the yearly average is less. We could indeed provide the number of months contributing to the aggregates, as supplementary material, and confirm what we just said. But we do not see how a user would benefit from that considering that we already described that over west and north China there are missing data. We also clearly state that over bright surface MODIS DB does better, as it was designed for this, and thus the use of MODIS data is recommended. We have looked at standard deviations and they are large over areas with high AOD and low over areas with low AOD, as expected. And highest values occur over areas most influenced by seasonal and / or interannual variations, which is the subject of paper 2. With statements on the performance of ATSR over bright surfaces, and the better performance of MODIS DBDT in Sect. 3.1.1 (3rd para), 4.1 (end 1st para), 4.3, Sect. 5 (end of 1st para) and Sect. 7 (end of 1st para), we don't see the need to stress this any further.

**Page 11 lines 8-15: the authors created a merged MODIS AOD product based on Collection 5.1 data, to compare with that provided in Collection 6 (as no official merge was provided in Collection 5.1). This is a nice thing to see as we as an algorithm team have not directly published C5.1/C6 difference maps from the merged product. However at present this part is not well integrated into the paper. The text says that "These differences will be further discussed after evaluation of the satellite AOD products versus available AERONET reference AOD data.", but I could not find the further discussion in the paper. Was this**

**removed in a draft, perhaps? If the authors want to show C5.1/C6 differences then my suggestion would be to add more discussion about how these changes may affect previously-published studies and their conclusions. Otherwise it can probably be deleted since C5.1 is considered obsolete now.**

Thanks for catching this, indeed we do not provide a detailed discussion of C6 results vs those from using C5.1. We have changed that sentence to "These differences need to be taken into account when comparing the results from using C6, as used in the current paper, with those in earlier papers using C5.1 data."

**Section 3.1.3: this section feels a bit disjointed from the rest. For example, why do we only have the thin strip of CALIPSO data from 35-45 N in Figure 5 rather than the whole study area? I think that better use could be made of CALIPSO's strengths here as well. For example it's good to see the seasonal vertical profile of extinction in Figure 6 across this strip, but showing something like depolarisation ratio or colour ratio or even aerosol type frequency would allow us to see the vertical structure and changing seasonal nature of aerosol composition, which we can't see from the extinction profile alone. I might also consider moving this part to fit in with the seasonal discussion near Figure 11.**

In Sect. 3 we present an overview of data available. Here we show only part of the data because those for all China have been presented and analyzed in Proestakis et al. under review in ACPD (https://doi.org/10.5194/acp-2017-797). Indeed there the depolarization is used to show seasonal variation of dust aerosol and the contribution of DAOD to total AOD. Here we show this slice as an example of the complementary information from CALIOP. This slice was chosen because it encompasses the Taklimakan desert as well as the BTH area, i.e. the desert with high dust emission and the BTH region which is strongly influenced by both dust and anthropogenic aerosol.

**Section 4: I like the validation shown, and the comparison between MODIS and ADV when both are collocated with AERONET in Figure 10. It would be good to add some statistics to Figure 10, similar to the other panels. Since the AERONET sites cover a range of aerosol and surface conditions, I think it would also be instructive to further separate the points in these figures to different regimes (e.g. compare dust/pollution dominated scenes, classified using AERONET AOD and Ångström exponent, maybe split sites by surface type). There is clearly variation in surface cover and aerosol type, and we know that algorithm errors are dependent on those factors, so it makes sense to look at the data that way. For example, in Figures 7 and 8 there are some high-AOD points with high biases and low biases. It may be that high are associated with one aerosol type and low another, or it may be that the data are unbiased but noisy. With the analysis as presented, we don't know. Understanding if there are systematic surface/aerosol type dependent errors in the ATSR and MODIS data here would also help understand the differences in seasonal and multiannual aggregates presented in the paper and then aid in the interpretation of the follow on part 2 paper. (I realise, and the authors state, that most of the AERONET sites are in low-lying veg-etated/suburban/urban areas, rather than deserts, so splitting by surface type may not be feasible. But doing a simple cut based on AERONET AOD and Ångström expo- nent to separate into background, fine-mode dominated, and coarse-mode dominated aerosols should be possible.)**

We thank the reviewer for the suggestion to add some statistics and have replaced the Figure by one including these. The other suggestion to look at effects of surface and aerosol types is very interesting but would require a separate study which is out of the scope of the current paper. This paper is not about understanding differences between MODIS and ATSR algorithms. The only reason why we evaluated the ATSR and MODIS AOD, with AERONET AOD as reference, was to see whether we can use the MODIS data to see how AOD varies after we lost AATSR.

**Section 5: although most of the seasonal analysis is to come in the follow up part 2 paper, some of it is included here. Figure 11 is very interesting (I saw also the on- line comment containing the correct seasonal panel ordering) as it reveals the large seasonality, going back to my earlier comment about multiannual aggregates alone not being useful. I think this figure though is also illustrative of my earlier point about needing to understand differences rather than just state them. Clearly both sensors see similar patterns and seasonality but there are offsets (and again the differences in C6.1 are relevant here). Looking at type-dependence of errors vs. AERONET (suggested above) helps answer this question. Adding different measurement types also helps. Right now we have ATSR (dual-view narrow swath) and MODIS (single-**

**view broad swath). I think the authors should move the CALIPSO seasonal maps (currently Figure 5) to be an extra column of this Figure, and expand them to cover the whole spatial domain. That provides another different measurement technique (active curtain profiling by lidar). Adding MISR as a fourth would also help: it is multi-view like AATSR, but has more angles (9 vs. 2), and a different spectral range, which provides different information content. Put them all together and you have a nice ensemble of 4 different seasonal climatologies from different sensors and retrieval techniques, and you will really begin to see where the data are consistent and where they are not. That tells data users where to be cautious and data producers where to focus effort.**

Indeed, this is a very interesting study, but as discussed above, CALIOP is presented in a different paper (Proestakis et al., ACPD in review) and the use of MISR goes beyond the scope of the current paper. Doing all this would likely requires another paper to write it all up.

**Figure 11: there are some strange artefacts near the northern end of the MODIS DJF panel (appears as SON in the Discussion paper). Are these really areas of zero AOD, or are these areas without valid retrievals, which should then appear white for missing data? I think it is more likely this is missing data; this should be checked. There's a similar smaller artefact along the Himalayas in the MODIS MAM/JJA panels (appear as DJF/MAM in the Discussion papers), near 35 N 80 E.**

We have looked at this in detail and are grateful that the reviewer caught this. Indeed we had zeros in our data files where there should have been NANs. In the corrected Fig. these pixels are not coloured.

**Section 6: I have mixed feelings about this section. I understand that temporal variations are one of the main topics of the follow on part 2 study by Sogacheva et al. Here the authors show that MODIS and ATSR time series are offset but have similar inter-annual variations. However these are annual aggregates across the whole of China so again the question is raised of how meaningful these numbers are (they are not representative of any one place in China or season so it is not clear what I would want to use these numbers to assess or calculate). In my view including it here opens a bit of a can of worms, while it is interesting to see. My suggestion is to remove Section 6 and save all discussion of time series and trends for the follow on paper. It is already fairly well signposted that seasonality and trends are the topics for the follow on, so I don't know that it is necessary to introduce that topic here. The end of the conclusion can be cut down to do this signposting as well.**

We do not share this point of view of the reviewer. We present two decades of AOD data, and the main purpose of the comparisons in this paper is to see whether indeed we can do this. However, we realize, and stress this again, that we do **not** discuss trends, only variations of yearly averaged AOD values and discuss that they change in time. We realize, as indicated, that there may be several reasons for this, including meteorological variations and large scale systems, as well as policy. This is common practice as done, e.g. by Zhao et al. (2017) and others referenced in this paper. Indeed we would open a can of worms if we tried to do quantitative trend analysis, but we don't do that here.

**Section 8: I think Sections 7 and 8 can be condensed into a single discussion and conclusion section. I don't see what is gained by having the separate section 8 since it is short, and most of what is in section 7 is fitting for a conclusion. The two are even somewhat repetitive, e.g. last sections mentioning time series will be discussed in Sogacheva et al. Condensing this down, as well as saving Section 6 for the follow-on, will help prevent the paper from getting too long if the analysis is expanded to cover some of my suggestions.**

We disagree with the reviewer. As shown on the ACP website, a separate conclusions section is recommended:
https://www.atmospheric-chemistry-and-physics.net/for_authors/manuscript_preparation.html
Furthermore, we feel that it is nice to have the conclusions from a study summarized at the end.

**Please also note the supplement to this comment:**
**https://www.atmos-chem-phys-discuss.net/acp-2017-838/acp-2017-838-RC1-supplement.pdf**
* * *
**Interactive comment on Atmos. Chem. Phys. Discuss., https://doi.org/10.5194/acp-2017-**

838, 2017.

---

## Author Comment (AC4) · 18 Dec 2017

**De Leeuw et al. reported a new 3-D climatology of AOD over mainland China based on the combination of MODIS, ATSR and AERONET. Compared with MODIS, the radiometer of ATSR extends the AOD time series albeit the large offset between these two products. This topic is not new, but the methods used are robust and analysis is sound. For instance, ground- based AERONET AODs have been used to validate the space-borne AOD, in addition to the intercomparisons between MODIS AOD and ATSR AOD. Overall, the paper is well written, and deserves publications in ACP if the following concerns being successfully addressed.**

We thank the anonymous reviewer for the careful consideration of the MS and the positive recommendation

**General comments:**

1. **Table 1: It is well known that the time period when AERONET stations operated in the past varies a lot by regions. The time period for each site in Table is suggested to be added so that the readers can be better informed. Likewise, the various time periods or AREONET sites can be interpreted by differentiating the scatters in Figs. 7 and 8 in various shapes. This will help give more information with regard to the validation results.**

    We have added the time periods of operation in Table, where we discriminated between periods when disruptions were 3 months or longer. However, in Figs 7 and 8 we have not differentiated between different stations since we do not used that kind of analysis. The study of the reasons for the differences between ATSR and MODIS is out of scope for the current paper and would make it too long. However, this would be interesting for a separate study warranting another paper (see also response to Rev#1 t).

2. **Why only CALIOP AOD over north parts of mainland China is shown in Fig. 5? The topic is the AOD climatology over China, and it will be useful to show CALIOP AOD throughout the study area. Meanwhile, you can highlight in Fig. 5 the region of interest (35-45 N; 70-150E) for the cross-section map in Fig.6.**

    In Sect. 3 we present an overview of data available. Here we show only part of the data because those for all China have been presented and analyzed in Proestakis et al. under review in ACPD (https://doi.org/10.5194/acp-2017-797). Here we show this slice as an example of the complementary information from CALIOP. This slice was chosen because it encompasses the Taklimakan desert as well as the BTH area,

i.e. the desert with high dust emission and the BTH region which is strongly influenced by both dust and anthropogenic aerosol. The area shown in Fig. 5 and Fig. 6 is the same.

**Minor comments:**

1. **Page 2, line 22-23: The following references could be considered to be added to inform the readers of the background of aerosol effect. "Aerosol particles are important because    of their effects on weather and climate (e.g., Rosenfeld et al., 2008; Koren et al., 2014; Guo et al., 2016), health (Pope et al., 2009; Anenberg et al., 2010), atmospheric chemistry, visibility(Sisler and Malm, 1994), cultural heritage, etc."**

**References:**
**Anenberg, SC, Horowitz, LW and Tong, DQ *et al*. An estimate of the global burden of anthropogenic ozone and fine particulate matter on premature human mortality using atmospheric modeling. Environmental Health Perspectives 2010; 118:1189-1195.**
**Guo, J., M. Deng, S. S. Lee, F. Wang, Z. Li, P. Zhai, H. Liu, W. Lv, W. Yao, and X. Li (2016), Delaying precipitation and lightning by air pollution over the Pearl River Delta. Part I: Observational analyses,    J.    Geophys.    Res.    Atmos.,    121,    6472–6488, doi:10.1002/2015JD023257.**
**Koren, I., G. Dagan, and O. Altaratz, 2014. From aerosol-limited to invigoration of warm    convective    clouds,    Science,    344(6188),    1143–1146, doi:10.1126/science.1252595.**
**Pope C. A., M. Ezzati, and D.W. Dockery. 2009. Fine-particulate air pollution and life expectancy in the United States. N. Engl. J. Med. 360: 376-386.**
**Rosenfeld, D., U. Lohmann, G. B. Raga, C. D. O'Dowd, M. Kulmala, S. Fuzzi, A. Reissell, and M. O. Andreae (2008), Flood or drought: How do aerosols affect precipitation?, Science, 321(5894), 1309–1313.**
**Sisler, J. F., and Malm, W. C. (1994). The relative importance of soluble aerosols to spatial and seasonal trends of impaired visibility in the United States. Atmospheric environment, 28(5), 851-862.**

These references have been added

2. **Page 17, line 14: "Varying between -20 and 80 minutes"? Also, the discussion as regards the overestimation of MODIS AOD relative to ATSR AOD could be problematic. In the morning, the boundary layer height (BLH) generally increases as a result of increasing turbulence caused by increasing incoming solar radiation (Guo et al., 2016; Stull et al., 1988; Petäjä et al., 2016). The later overpass time of MODIS/Terra will have higher BLHs, resulting in lower aerosol concentrations and smaller AOD. This is opposite to the results shown in Fig.10. Therefore, more insightful discussion is much needed here (Li et al., 2017).**

The difference in overpass time, with possible higher BL due to increasing turbulence mixing, was exactly the reason for showing Fig. 10, as discussed in the paragraph above the Fig. However, we do not see a systematic difference related to difference in overpass time indicated with the colours. The colour scale extends to -20 to avoid excluding data points where the difference is small or even negative. As shown in the Fig. there are some data points with a difference in overpass time of 0-20 minutes. Note that the equator passing times differ by 30 min, but the actually overpass time, which can be evaluated from the data files, may be different from 30 min due to swath width.

**References:**

Guo, J., Miao, Y., Zhang, Y., Liu, H., Li, Z., Zhang, W., He, J., Lou, M., Yan, Y., Bian, L., and Zhai, P.: The climatology of planetary boundary layer height in China derived from radiosonde and reanalysis data, Atmos. Chem. Phys., 16, 13309-13319, doi:10.5194/acp- 16-13309-2016, 2016.

Li Z., J. Guo, A. Ding, H. Liao, J. Liu, Y. Sun, T. Wang, H. Xue, H. Zhang, B. Zhu, 2017. Aerosol and boundary-layer interactions and impact on air quality, National Science Review, doi: 10.1093/nsr/nwx117.

Petäjä, T, Jarvi, L and Kerminen, VM et al. Enhanced air pollution via aerosol-boundary layer feedback in China. Scientific Reports 2016; 6: 18998.

Stull, RB. An Introduction to Boundary Layer Meteorology, edited by R. B. Stull, Springer Netherlands, Dordrecht 1988.

---

## Author Comment (AC5) · 18 Dec 2017

[revised manuscript text omitted]

**Page 35: [3] Formatted**       **Gerrit de Leeuw**       **12/14/2017 6:53:00 PM**

Level 1, Space Before:  24 pt, After:  12 pt, Line spacing:  single, Keep with next, Keep lines together

---

## Author Comment (AC6) · 18 Dec 2017

In the tracked version uploaded a few hours ago, some Figures were not replaced. This affects the changes in the MODIS C6 NaN , which were earlier plotted as zero (o). This has now also been corrected. We also replaced ATSR figures. Please use this version to see the changes in the MS revision2

Please also note the supplement to this comment:
https://www.atmos-chem-phys-discuss.net/acp-2017-838/acp-2017-838-AC6-supplement.pdf

2017.

**Supplement:**

[revised manuscript text omitted]

Greek

| Page 26: [2] Formatted | Gerrit de Leeuw | 12/18/2017 4:23:00 PM |
| --- | --- | --- |

Greek

Greek

| Page 26: [2] Formatted | Gerrit de Leeuw | 12/18/2017 4:23:00 PM |
|---|---|---|

Greek

| Page 26: [2] Formatted | Gerrit de Leeuw | 12/18/2017 4:23:00 PM |
|---|---|---|

Greek

| Page 26: [2] Formatted | Gerrit de Leeuw | 12/18/2017 4:23:00 PM |
|---|---|---|

Greek

| Page 26: [2] Formatted | Gerrit de Leeuw | 12/18/2017 4:23:00 PM |
|---|---|---|

Greek

| Page 26: [2] Formatted | Gerrit de Leeuw | 12/18/2017 4:23:00 PM |
|---|---|---|

Greek

| Page 26: [2] Formatted | Gerrit de Leeuw | 12/18/2017 4:23:00 PM |
|---|---|---|

Greek

| Page 26: [2] Formatted | Gerrit de Leeuw | 12/18/2017 4:23:00 PM |
|---|---|---|

Greek

| Page 26: [2] Formatted | Gerrit de Leeuw | 12/18/2017 4:23:00 PM |
|---|---|---|

Greek

| Page 26: [2] Formatted | Gerrit de Leeuw | 12/18/2017 4:23:00 PM |
|---|---|---|

Greek

| Page 26: [2] Formatted | Gerrit de Leeuw | 12/18/2017 4:23:00 PM |
|---|---|---|

Greek

| Page 26: [2] Formatted | Gerrit de Leeuw | 12/18/2017 4:23:00 PM |
|---|---|---|

Greek

| Page 26: [2] Formatted | Gerrit de Leeuw | 12/18/2017 4:23:00 PM |
|---|---|---|

Greek

| Page 26: [2] Formatted | Gerrit de Leeuw | 12/18/2017 4:23:00 PM |
|---|---|---|

Greek

**Page 26: [2] Formatted** | **Gerrit de Leeuw** | **12/18/2017 4:23:00 PM**

Greek

**Page 26: [2] Formatted** | **Gerrit de Leeuw** | **12/18/2017 4:23:00 PM**

Greek

**Page 26: [2] Formatted** | **Gerrit de Leeuw** | **12/18/2017 4:23:00 PM**

Greek

**Page 26: [2] Formatted** | **Gerrit de Leeuw** | **12/18/2017 4:23:00 PM**

Greek

**Page 26: [2] Formatted** | **Gerrit de Leeuw** | **12/18/2017 4:23:00 PM**

Greek

**Page 26: [2] Formatted** | **Gerrit de Leeuw** | **12/18/2017 4:23:00 PM**

Greek

**Page 26: [2] Formatted** | **Gerrit de Leeuw** | **12/18/2017 4:23:00 PM**

Greek

**Page 26: [2] Formatted** | **Gerrit de Leeuw** | **12/18/2017 4:23:00 PM**

Greek

**Page 26: [2] Formatted** | **Gerrit de Leeuw** | **12/18/2017 4:23:00 PM**

Greek

**Page 26: [2] Formatted** | **Gerrit de Leeuw** | **12/18/2017 4:23:00 PM**

Greek

**Page 26: [2] Formatted** | **Gerrit de Leeuw** | **12/18/2017 4:23:00 PM**

Greek

**Page 26: [2] Formatted** | **Gerrit de Leeuw** | **12/18/2017 4:23:00 PM**

Greek

**Page 26: [2] Formatted** | **Gerrit de Leeuw** | **12/18/2017 4:23:00 PM**

Greek

**Page 26: [2] Formatted** | **Gerrit de Leeuw** | **12/18/2017 4:23:00 PM**

Greek

| Page 26: [2] Formatted | Gerrit de Leeuw | 12/18/2017 4:23:00 PM |
|---|---|---|

Greek

| Page 26: [2] Formatted | Gerrit de Leeuw | 12/18/2017 4:23:00 PM |
|---|---|---|

Greek

| Page 26: [2] Formatted | Gerrit de Leeuw | 12/18/2017 4:23:00 PM |
|---|---|---|

Greek

| Page 26: [2] Formatted | Gerrit de Leeuw | 12/18/2017 4:23:00 PM |
|---|---|---|

Greek

| Page 26: [2] Formatted | Gerrit de Leeuw | 12/18/2017 4:23:00 PM |
|---|---|---|

Greek

| Page 26: [2] Formatted | Gerrit de Leeuw | 12/18/2017 4:23:00 PM |
|---|---|---|

Greek

| Page 26: [2] Formatted | Gerrit de Leeuw | 12/18/2017 4:23:00 PM |
|---|---|---|

Greek

| Page 26: [2] Formatted | Gerrit de Leeuw | 12/18/2017 4:23:00 PM |
|---|---|---|

Greek

| Page 26: [2] Formatted | Gerrit de Leeuw | 12/18/2017 4:23:00 PM |
|---|---|---|

Greek

| Page 26: [2] Formatted | Gerrit de Leeuw | 12/18/2017 4:23:00 PM |
|---|---|---|

Greek

| Page 26: [2] Formatted | Gerrit de Leeuw | 12/18/2017 4:23:00 PM |
|---|---|---|

Greek

| Page 26: [2] Formatted | Gerrit de Leeuw | 12/18/2017 4:23:00 PM |
|---|---|---|

Greek

| Page 26: [2] Formatted | Gerrit de Leeuw | 12/18/2017 4:23:00 PM |
|---|---|---|

Greek

| Page 26: [2] Formatted | Gerrit de Leeuw | 12/18/2017 4:23:00 PM |
|---|---|---|

Greek

| Page 26: [2] Formatted | Gerrit de Leeuw | 12/18/2017 4:23:00 PM |
|---|---|---|

Greek

| Page 26: [2] Formatted | Gerrit de Leeuw | 12/18/2017 4:23:00 PM |
|---|---|---|

Greek

| Page 26: [2] Formatted | Gerrit de Leeuw | 12/18/2017 4:23:00 PM |
|---|---|---|

Greek

| Page 26: [2] Formatted | Gerrit de Leeuw | 12/18/2017 4:23:00 PM |
|---|---|---|

Greek

| Page 26: [2] Formatted | Gerrit de Leeuw | 12/18/2017 4:23:00 PM |
|---|---|---|

Greek

| Page 26: [2] Formatted | Gerrit de Leeuw | 12/18/2017 4:23:00 PM |
|---|---|---|

Greek

| Page 26: [2] Formatted | Gerrit de Leeuw | 12/18/2017 4:23:00 PM |
|---|---|---|

Greek

| Page 26: [2] Formatted | Gerrit de Leeuw | 12/18/2017 4:23:00 PM |
|---|---|---|

Greek

| Page 26: [2] Formatted | Gerrit de Leeuw | 12/18/2017 4:23:00 PM |
|---|---|---|

Greek

| Page 26: [2] Formatted | Gerrit de Leeuw | 12/18/2017 4:23:00 PM |
|---|---|---|

Greek

| Page 35: [3] Deleted | Gerrit de Leeuw | 12/14/2017 6:53:00 PM |
|---|---|---|

**Figures and Tables**

[revised manuscript text omitted]

**Page 35: [4] Formatted**          **Gerrit de Leeuw**          **12/14/2017 6:53:00 PM**

Level 1, Space Before: 24 pt, After: 12 pt, Line spacing: single, Keep with next, Keep lines together

---

## Author Response (AR1)

Dear Editor, Dear Stelios,

We thank the referees and others who commented on our initial manuscript for their valuable suggestions which certainly improved the MS. Responses to short comments have been uploaded soon after these comments were received and suggestions were followed up as mentioned in these responses. Responses to Ref#3 and 4, who is the same referee, were provided in the submission phase and responses to Ref# 1 and 2 were uploaded earlier this week. We have changed the MS following our responses, although for a number of these comments we considred that no changes were needed. Below please find the revised MS with changes tracked. A clean version has also been uploaded.

Best regards, Merry Christmas and Happy New Year !

Gerrit

[revised manuscript text omitted]

Greek

Greek

| **Page 27: [2] Formatted** | **Gerrit de Leeuw** | **12/18/2017 4:23:00 PM** |

Greek

| **Page 27: [2] Formatted** | **Gerrit de Leeuw** | **12/18/2017 4:23:00 PM** |

Greek

| **Page 27: [2] Formatted** | **Gerrit de Leeuw** | **12/18/2017 4:23:00 PM** |

Greek

| **Page 27: [2] Formatted** | **Gerrit de Leeuw** | **12/18/2017 4:23:00 PM** |

Greek

| **Page 27: [2] Formatted** | **Gerrit de Leeuw** | **12/18/2017 4:23:00 PM** |

Greek

| **Page 27: [2] Formatted** | **Gerrit de Leeuw** | **12/18/2017 4:23:00 PM** |

Greek

| **Page 27: [2] Formatted** | **Gerrit de Leeuw** | **12/18/2017 4:23:00 PM** |

Greek

| **Page 27: [2] Formatted** | **Gerrit de Leeuw** | **12/18/2017 4:23:00 PM** |

Greek

| **Page 27: [2] Formatted** | **Gerrit de Leeuw** | **12/18/2017 4:23:00 PM** |

Greek

| **Page 27: [2] Formatted** | **Gerrit de Leeuw** | **12/18/2017 4:23:00 PM** |

Greek

| **Page 27: [2] Formatted** | **Gerrit de Leeuw** | **12/18/2017 4:23:00 PM** |

Greek

| **Page 27: [2] Formatted** | **Gerrit de Leeuw** | **12/18/2017 4:23:00 PM** |

Greek

| **Page 27: [2] Formatted** | **Gerrit de Leeuw** | **12/18/2017 4:23:00 PM** |

Greek

| **Page 27: [2] Formatted** | **Gerrit de Leeuw** | **12/18/2017 4:23:00 PM** |

Greek

| **Page 27: [2] Formatted** | **Gerrit de Leeuw** | **12/18/2017 4:23:00 PM** |

Greek

| **Page 27: [2] Formatted** | **Gerrit de Leeuw** | **12/18/2017 4:23:00 PM** |

Greek

| **Page 27: [2] Formatted** | **Gerrit de Leeuw** | **12/18/2017 4:23:00 PM** |

Greek

| **Page 27: [2] Formatted** | **Gerrit de Leeuw** | **12/18/2017 4:23:00 PM** |

Greek

| **Page 27: [2] Formatted** | **Gerrit de Leeuw** | **12/18/2017 4:23:00 PM** |

Greek

| **Page 27: [2] Formatted** | **Gerrit de Leeuw** | **12/18/2017 4:23:00 PM** |

Greek

| **Page 27: [2] Formatted** | **Gerrit de Leeuw** | **12/18/2017 4:23:00 PM** |

Greek

| **Page 27: [2] Formatted** | **Gerrit de Leeuw** | **12/18/2017 4:23:00 PM** |

Greek

| **Page 27: [2] Formatted** | **Gerrit de Leeuw** | **12/18/2017 4:23:00 PM** |

Greek

| **Page 27: [2] Formatted** | **Gerrit de Leeuw** | **12/18/2017 4:23:00 PM** |

Greek

| **Page 27: [2] Formatted** | **Gerrit de Leeuw** | **12/18/2017 4:23:00 PM** |

Greek

| **Page 27: [2] Formatted** | **Gerrit de Leeuw** | **12/18/2017 4:23:00 PM** |

Greek

| **Page 27: [2] Formatted** | **Gerrit de Leeuw** | **12/18/2017 4:23:00 PM** |

Greek

| **Page 27: [2] Formatted** | **Gerrit de Leeuw** | **12/18/2017 4:23:00 PM** |

Greek

**Page 27: [2] Formatted**          **Gerrit de Leeuw**          **12/18/2017 4:23:00 PM**

Greek

| Page 27: [2] Formatted | Gerrit de Leeuw | 12/18/2017 4:23:00 PM |
|---|---|---|

Greek

| Page 27: [2] Formatted | Gerrit de Leeuw | 12/18/2017 4:23:00 PM |
|---|---|---|

Greek

| Page 27: [2] Formatted | Gerrit de Leeuw | 12/18/2017 4:23:00 PM |
|---|---|---|

Greek

| Page 27: [2] Formatted | Gerrit de Leeuw | 12/18/2017 4:23:00 PM |
|---|---|---|

Greek

| Page 27: [2] Formatted | Gerrit de Leeuw | 12/18/2017 4:23:00 PM |
|---|---|---|

Greek

| Page 27: [2] Formatted | Gerrit de Leeuw | 12/18/2017 4:23:00 PM |
|---|---|---|

Greek

| Page 27: [2] Formatted | Gerrit de Leeuw | 12/18/2017 4:23:00 PM |
|---|---|---|

Greek

| Page 27: [2] Formatted | Gerrit de Leeuw | 12/18/2017 4:23:00 PM |
|---|---|---|

Greek

| Page 27: [2] Formatted | Gerrit de Leeuw | 12/18/2017 4:23:00 PM |
|---|---|---|

Greek

| Page 27: [2] Formatted | Gerrit de Leeuw | 12/18/2017 4:23:00 PM |
|---|---|---|

Greek

| Page 27: [2] Formatted | Gerrit de Leeuw | 12/18/2017 4:23:00 PM |
|---|---|---|

Greek

| Page 27: [2] Formatted | Gerrit de Leeuw | 12/18/2017 4:23:00 PM |
|---|---|---|

Greek

| Page 27: [2] Formatted | Gerrit de Leeuw | 12/18/2017 4:23:00 PM |
|---|---|---|

Greek

| Page 27: [2] Formatted | Gerrit de Leeuw | 12/18/2017 4:23:00 PM |
|---|---|---|

Greek

| Page 27: [2] Formatted | Gerrit de Leeuw | 12/18/2017 4:23:00 PM |
|---|---|---|

Greek

| Page 27: [2] Formatted | Gerrit de Leeuw | 12/18/2017 4:23:00 PM |
|---|---|---|

Greek

| Page 27: [2] Formatted | Gerrit de Leeuw | 12/18/2017 4:23:00 PM |
|---|---|---|

Greek

| Page 27: [2] Formatted | Gerrit de Leeuw | 12/18/2017 4:23:00 PM |
|---|---|---|

Greek

| Page 27: [2] Formatted | Gerrit de Leeuw | 12/18/2017 4:23:00 PM |
|---|---|---|

Greek

| Page 27: [2] Formatted | Gerrit de Leeuw | 12/18/2017 4:23:00 PM |
|---|---|---|

Greek

| Page 27: [2] Formatted | Gerrit de Leeuw | 12/18/2017 4:23:00 PM |
|---|---|---|

Greek

| Page 27: [2] Formatted | Gerrit de Leeuw | 12/18/2017 4:23:00 PM |
|---|---|---|

Greek

| Page 27: [2] Formatted | Gerrit de Leeuw | 12/18/2017 4:23:00 PM |
|---|---|---|

Greek

| Page 27: [2] Formatted | Gerrit de Leeuw | 12/18/2017 4:23:00 PM |
|---|---|---|

Greek

| Page 36: [3] Deleted | Gerrit de Leeuw | 12/14/2017 6:53:00 PM |
|---|---|---|

**Figures and Tables**

[revised manuscript text omitted]

**Page 36: [4] Formatted**    **Gerrit de Leeuw**    **12/14/2017 6:53:00 PM**

Level 1, Space Before:  24 pt, After:  12 pt, Line spacing:  single, Keep with next, Keep lines together